



# Scaling procedure for straightforward computation of sorptivity

Laurent Lassabatere[1], Pierre-Emmanuel Peyneau[2], Deniz Yilmaz[3], Joseph Pollacco[4],
Jesús Fernández-Gálvez[5], Borja Latorre[6], David Moret-Fernández[6], Simone Di Prima[7],
Mehdi Rahmati[8,9], Ryan D. Stewart[10], Majdi Abou Najm[11], Claude Hammecker[12], and
Rafael Angulo-Jaramillo[1]

[1]Univ Lyon, Université Claude Bernard Lyon 1, CNRS, ENTPE, UMR5023 LEHNA, F-69518, Vaulx-en-Velin, France
[2]GERS-LEE, Univ Gustave Eiffel, IFSTTAR, F-44344 Bouguenais, France
[3]Civil Engineering Department, Engineering Faculty, Munzur University, Tunceli, Turkey
[4]Manaaki Whenua - Landcare Research, 7640 Lincoln, New Zealand
[5]Department of Regional Geographic Analysis and Physical Geography, University of Granada, Granada, 18071, Spain
[6]Departamento de Suelo y Agua, Estación Experimental de Aula Dei, Consejo Superior de Investigaciones Científicas
(CSIC), PO Box 13034, 50080 Zaragoza, Spain
[7]Agricultural Department, University of Sassari, Viale Italia, 39, 07100 Sassari, Italy
[8]Department of Soil Science and Engineering, Faculty of Agriculture, University of Maragheh, Maragheh, Iran
[9]Forschungszentrum Jülich GmbH, Institute of Bio- and Geosciences: Agrosphere (IBG-3), Jülich, Germany
[10]School of Plant and Environmental Sciences, Virginia Polytechnic Institute and State University, Blacksburg, VA, United
States
[11]Department of Land, Air and Water Resources, University of California, Davis, CA 95616, United States
[12]University of Montpellier, UMR LISAH, IRD, Montpellier, France

**Correspondence:** Laurent Lassabatere (laurent.lassabatere@entpe.fr)

**Abstract.**

Sorptivity is a parameter of primary importance in the study of unsaturated flow in soils. This integral parameter is often considered for modeling the computation of water infiltration into vertical soil profiles (1D or 3D axisymmetric geometry). Sorptivity can be directly estimated from the knowledge of the soil hydraulic functions (water retention of hydraulic conductivity), using the integral formulation of Parlange (Parlange, 1975). However, it requires the prior determination of the soil
hydraulic diffusivity and its numerical integration between the initial and the final saturation degrees, which may be tricky for some instances (e.g., coarse soil with diffusivity functions quasi-infinite close to saturation). In this paper, we present a specific scaling procedure for the computation of sorptivity considering slightly positive water pressure heads at the soil surface and initial dry conditions (corresponding to most water infiltration on the field). The square sorptivity is related to the
square dimensionless sorptivity (referred to as $c_p$ parameter) corresponding to a unit soil (i.e., unit values of all the scaled parameters and zero residual water content) utterly dry at the initial state and saturated at the final state. The $c_p$ parameter was computed numerically and analytically for five current models: delta functions (Green and Ampt model), Brooks and Corey, van Genuchten-Mualem, van Genuchten-Burdine, and Kosugi models as a function of the shape parameters. The values are tabulated and can be easily used to determine any dimensional sorptivity value for any case. We propose brand-new analytical
expressions for some of the models and validate previous formulations for the other models. Our numerical results also showed that the relation between the $c_p$ parameters and shape parameters strongly depends on the chosen model, with either increasing





or decreasing trends when moving from coarse to fine soils. These results highlight the need for carefully selecting the proper model for the description of the water retention and hydraulic conductivity functions for the rigorous estimation of sorptivity. Present results show the need to understand better the hydraulic model's mathematical properties, including the links between

their parameters, and, secondly, to better relate these properties to the physical processes of water infiltration into soils.

## 1   Introduction

Soil sorptivity represents the capacity of a soil to absorb or desorb liquid by capillarity, and is therefore one of the key factors for modelling water infiltration into soil (Cook and Minasny, 2011). Knowledge of soil sorptivity is also important when deciphering soil physical properties such as hydraulic conductivity from infiltration tests (e.g., Lassabatere et al., 2006). Sorptivity

is incorporated in a wide range of infiltration models (Angulo-Jaramillo et al., 2016; Lassabatere et al., 2009, 2014, 2019) and must be properly quantified from the soil hydraulic functions and the initial and surface hydric conditions. In this study, we address this issue and propose a new scaling procedure to simplify its computation.

One of the first equations proposed for the computation of sorptivity was developed by Philip (1957) for the modelling of 1D gravity-free water infiltration:

$$
\begin{cases}
I(t) = S\left(\theta_0, \theta_1\right)\sqrt{t} \\
S\left(\theta_0, \theta_1\right) = \int_{\theta_0}^{\theta_1} \chi\left(\theta\right) d\theta
\end{cases}
\tag{1}
$$

In the above equations, $S(\theta_0, \theta_1)$ stands for the sorptivity between $\theta_0$ and $\theta_1$, $\chi = \chi\left(\theta\right) = \frac{x(\theta)}{\sqrt{t}}$ stands for the Boltzmann transformation variable, $\theta_0$ is the initial water content, and $\theta_1$ the final water content corresponding also to the water content applied at the soil surface. The main shortcoming is that such a procedure requires to proceed to the numerical modeling of the horizontal infiltration, for given hydraulic functions and initial and final conditions. Then, the Boltzmann transformation must

be computed from the modeled water content profile and integrated between the initial and final water contents (see Eq. (1)). Such a procedure may be time-consuming and subject to numerical instabilities spoiled with numerical errors.

To avoid such a complexity, Parlange (1975) proposed a formulation that relates directly sorptivity to the hydraulic functions and the initial and final water contents:

$$
S_D^2\left(\theta_0, \theta_1\right) = \int_{\theta_0}^{\theta_1} \left(\theta_1 + \theta - 2\theta_0\right) D\left(\theta\right) d\theta
\tag{2}
$$

where $D(\theta) = K(\theta)dh/d\theta$ is the hydraulic diffusivity function. While the above equation provides the diffusivity form for sorptivity determination, it can be equally defined as a function of the hydraulic conductivity function, $K(h) = K\left(\theta\left(h\right)\right)$:

$$
\begin{aligned}
S_K^2\left(h_0, h_1\right) &= \int_{h_0}^{h_1} \left(\theta\left(h_1\right) + \theta\left(h\right) - 2\,\theta\left(h_0\right)\right) K\left(h\right) dh \\
&= \int_{h_0}^{h_1} \left(\theta_1 + \theta\left(h\right) - 2\,\theta_0\right) K\left(h\right) dh
\end{aligned}
\tag{3}
$$





where $h_0$ and $h_1$ are respectively the initial and final water pressure heads, and $\theta_0 = \theta(h_0)$ and $\theta_1 = \theta(h_1)$. For the sake of clarity, the functions $S_D^2$ and $S_K^2$ are respectively referred to as the "diffusivity" and "conductivity" forms of sorptivity. $S_D^2$ and

$S_K^2$ are equivalent so long as the water retention function $\theta(h)$ is bijective over the water pressure head interval $[h_0, h_1]$, which is the case when the water pressure head at surface is lower than the air-entry water pressure head, $h_1 \leq h_a$. Then, Eq. (3) can be easily deduced from Eq. (2) by a simple change of variable from $\theta$ to $h$:

$$
\begin{aligned}
S_K^2(h_0, h_1 \leq h_a) &= S_D^2(\theta(h_0), \theta(h_1)) \\
&= S_D^2(\theta_0, \theta_1)
\end{aligned}
\tag{4}
$$

Otherwise, when the surface water pressure head exceeds the soil air-entry pressure, i.e., $h_1 > h_a(\leq 0)$, sorptivity must

be computed using Eq. (3) (Ross et al., 1996). Indeed, the function $\theta(h)$ is no longer bijective over the interval $[h_0, h_1]$. Consequently, Eq. (2) and Eq. (3) are no longer equivalent. In this case, the integration involved in Eq. (3) must be divided into two parts, one part integrating over the interval $[h_0, h_a]$ ensuring the bijectivity of the function $\theta(h)$, and the other part retaining the integration interval $[h_a, h_1]$ corresponding to the saturated part of the integration (Ross et al., 1996). Then, the change of variable from $\theta$ to $h$ in the first integral leads to the retrieval of $S_D^2$:

$$
\begin{aligned}
S_K^2(h_0, h_1 > h_a) &= \int_{h_0}^{h_a} (\theta_1 + \theta(h) - \theta_0) K(\theta(h)) \, dh + \int_{h_a}^{h_1} (\theta_1 + \theta(h) - 2\theta_0) K(\theta(h)) \, dh \\
&= \int_{h_0}^{h_a} (\theta_1 + \theta(h) - 2\theta_0) K(\theta(h)) \, dh + 2(\theta_s - \theta_0) K_s \int_{h_a}^{h_1} dh \\
&= \int_{h_0}^{h_a} (\theta_1 + \theta(h) - 2\theta_0) K(\theta(h)) \, dh + 2(\theta_s - \theta_0) K_s (h_1 - h_a) \\
&= \int_{\theta_0}^{\theta_s} (\theta_1 + \theta - 2\theta_0) D(\theta) \, d\hat{I}_s + 2(\theta_s - \theta_0) K_s (h_1 - h_a) \\
&= S_D^2(\theta_0, \theta_s) + 2(\theta_s - \theta_0) K_s (h_1 - h_a)
\end{aligned}
\tag{5}
$$

The computation of sorptivity with Eqs. (2) or (3) requires the choice of a set of hydraulic functions from a wide range of available models for the characterization of soils. Here, we consider five of the most widely used hydraulic functions. Firstly, the Brooks and Corey (1964) model, referred to as the "BC" model, is among the first hydraulic functions of soil physics (Hillel). The BC model involves power functions for both the water retention (WR) and hydraulic conductivity (HC) functions,

thus allowing for analytical integration of Eq. (2) and leading to analytical expressions for sorptivity (e.g. Varado et al., 2006). Secondly, the van Genuchten – Burdine (vGB) model that combines van Genuchten (1980) model with Burdine conditions for water retention (WR) and the Brooks and Corey (1964) model for Hydraulic Conductivity (HC). The vGB model has been used for the development of BEST methods and for the characterization of soil hydraulic properties (Lassabatere et al., 2006; Yilmaz et al., 2010; Bagarello et al., 2014). Thirdly, the van Genuchten – Mualem (vGM) model that combines van Genuchten





(1980) model with Mualem's condition for WR and the Mualem (1976) capillary model for HC. The vGM is among the most

widely-used models and often used for the numerical modelling of flow in the vadose zone (Šimůnek et al., 2003). Fourthly,

the Kosugi (KG) model (Kosugi, 1996), is quite popular since it relates the water retention function to physical characteristics

of the soil pore size distribution assuming log-normal distributions. Finally, the Delta model (d), corresponding to the Dirac

delta functions, i.e., the use of stepwise functions for the description of WR and HC, was added to the list of studied models.

Indeed, this model is often considered for the analytical resolution of Richards equations and the determination of analytical

expressions for water infiltration, like the Green and Ampt approach (Triadis and Broadbridge, 2012).

These five models have the following mathematical expressions (Triadis and Broadbridge, 2012; Brooks and Corey, 1964;

van Genuchten, 1980; Mualem, 1976; Kosugi, 1996):

$$
\begin{cases}
\theta_d(h) = \begin{cases} \theta_s & h \geq h_d \\ \theta_r & h < h_d \end{cases} \\[2ex]
K_d(h) = \begin{cases} K_s & h \geq h_d \\ 0 & h < h_d \end{cases}
\end{cases}
\qquad \text{Delta model} \qquad (6)
$$


$$
\begin{cases}
\theta_{BC}(h) = \begin{cases} \theta_s & h \geq h_{BC} \\ \theta_r + (\theta_s - \theta_r)\left(\frac{h_{BC}}{h}\right)^{\lambda_{BC}} & h < h_{BC} \end{cases} \\[2ex]
K_{BC}(\theta) = K_s \left(\frac{\theta - \theta_r}{\theta_s - \theta_r}\right)^{\eta_{BC}}
\end{cases}
\qquad \text{BC model} \qquad (7)
$$

$$
\begin{cases}
\theta_{vGB}(h) = \theta_r + (\theta_s - \theta_r)\left(1 + \frac{h}{h_{vGB}}\right)^{-m_{vGB}} \\[2ex]
K_{vGB}(\theta) = K_s \left(\frac{\theta - \theta_r}{\theta_s - \theta_r}\right)^{\eta_{vGB}}
\end{cases}
\qquad m_{vGB} = 1 - \frac{2}{n_{vGB}}
\qquad \text{vGB model} \qquad (8)
$$

$$
\begin{cases}
\theta_{vGM}(h) = \theta_r + (\theta_s - \theta_r)\left(1 + \frac{h}{h_{vGM}}\right)^{-m_{vGM}} \\[2ex]
K_{vGM}(\theta) = K_s \left(\frac{\theta - \theta_r}{\theta_s - \theta_r}\right)^{l_{vGM}} \left(1 - \left(1 - \left(\frac{\theta - \theta_r}{\theta_s - \theta_r}\right)^{\frac{1}{m_{vGM}}}\right)^{m_{vGM}}\right)^2
\end{cases}
\qquad m_{vGM} = 1 - \frac{1}{n_{vGM}}
\qquad \text{vGM model} \qquad (9)
$$

$$
\begin{cases}
\theta_{KG}(h) = \theta_r + (\theta_s - \theta_r)\, erfc\left(\frac{(ln(h) - ln(h_{KG}))}{\sqrt{2}\sigma_{KG}}\right) \\[2ex]
K_{KG}(\theta) = K_s \left(\frac{\theta - \theta_r}{\theta_s - \theta_r}\right)^{l_{KG}} \left(\frac{1}{2} erfc\left(erfc^{-1}\left(2\frac{\theta - \theta_r}{\theta_s - \theta_r}\right) + \frac{\sigma_{KG}}{2}\right)\right)^2
\end{cases}
\qquad \text{KG model} \qquad (10)
$$

Where $H$ stands for the one-sided Heaviside step function: $H(x < 0) = 0$, $H(x \geq 0) = 1$ (Triadis and Broadbridge, 2012);

$erfc$ stands for the complementary error function. These models involve several specific hydraulic shape parameters and the

following common scale hydraulic parameters: residual water content, $\theta_r$, saturated water content, $\theta_s$, scale parameter for the

water pressure head, $h_g$, (=$h_d$,$h_{BC}$,$h_{vGB}$,$h_{vGM}$, or $h_{KG}$), and saturated hydraulic conductivity, $K_s$. The Delta and BC models





involve a non-null air-entry water pressure head, $h_d$ and $h_{BC}$, meaning that air needs a given suction to enter into the soil and to desaturate the soil. For the sake of simplicity, the scale parameter for water pressure head is fixed at the air-entry pressure head, i.e., $h_g = h_d$ and $h_g = h_{BC}$, respectively.

The computation of sorptivity by applying the Eqs. (2) or (3) to hydraulic models, and in particular to those selected for this study, i.e., Eqs. (6)-(10), are quite tricky, given the complexity of the hydraulic functions. Such computation might exhibit the following shortcomings. First of all, the diffusivity functions must be determined analytically, by multiplying the hydraulic conductivity with the derivative of water pressure head with regards to water content, which may involve complex algebraic operations. Then, the integration involved in the right-hand side of Eq. (3) may lead to numerical indetermination for very low

initial water pressure heads, in case of very dry initial conditions. Meanwhile, the integration involved in the right-hand side of Eq. (2) may pose numerical shortcomings for infinite hydraulic diffusivity, which is the case of some of the hydraulic functions detailed above, Eqs. (6)-(10).

In this study, we propose a specific scaling procedure to avoid all these shortcomings and to simplify the computation of sorptivity for the hydraulic models described in Eqs. (6)-(10) under the boundary conditions of a slightly positive water pres-

sure head at surface and relatively dry initial conditions. We focus on these conditions since they constitute the most common experimental conditions for most water infiltration experiments and related procedures for characterizing soil hydraulic properties (Angulo-Jaramillo et al., 2016). In particular, these conditions feature the Beerkan method that involves pouring water into a ring placed on the ground (Braud, 2005; Lassabatere et al., 2006). The theory section details the scaling procedure that relates the square sorptivity to the square scaled sorptivity, $c_p = S_K^{*2}(-\infty, 0)$, the product of scale parameters, and correcting

factors accounting for the contribution of initial hydric conditions. The square scaled sorptivity corresponds to the sorptivity of a unit soil (unit value for all the scale parameters, except the residual water content fixed at zero) and for the whole range of water pressure head, i.e., $(-\infty, 0]$. It depends only on the soil hydraulic shape parameters, and its determination features the main algebraic complexity of the whole scaling procedure, the rest relies on simple algebraic operations (multiplication and sums). It was then computed for the hydraulic models defined by Eqs. (6)-(10), either analytically when feasible or numeri-

cally, otherwise. For each model, it was computed and tabulated as a function of a shape index that characterizes the spreads of the water retention function (from gradual to stepwise shapes, corresponding to soils with a broad or a very narrow pore size distribution, respectively). The evolution of the square scaled sorptivity versus the shape index was compared and discussed between models. In the last section, we illustrate the application of the proposed scaling procedure. We show how the tabulated values of the square scaled sorptivity $c_p$ can be used to upscale sorptivity and provide easily the sorptivity corresponding to

zero water pressure head at surface any relatively small initial water content.

## 2   Theory

### 2.1   Global scaling procedure

The global scaling procedures rely on several steps emanating from previous studies. Scaling has often been used to separate the effects of sorptivity drivers and simplify the computation of sorptivity. Ross et al. (1996) suggested to scale water content,





water pressure head and hydraulic conductivity using the following equations:

$$
\begin{cases}
S_e = \frac{\theta - \theta_r}{\theta_s - \theta_r} \\
h^* = \frac{h}{|h_g|} \\
K_r = \frac{K}{K_s}
\end{cases}
\tag{11}
$$

This scaling procedure defines the dimensionless water retention, $S_e(h^*)$, the dimensionless (or relative) hydraulic conductivity, $K_r(S_e)$, and the dimensionless hydraulic diffusivity function $D^*(S_e) = K_r(S_e)\frac{dh^*}{dS_e}$. These dimensionless hydraulic functions define the hydraulic characteristics of the unit soil that has the same values for the shape parameters and unit value

for all the scale parameters, $\theta_s = 1$, $h_g = 1$, $K_s = 1$, excepted the residual water content that is fixed at zero. The use of the scaling Eqs. (11) allows us to relate the square sorptivity (dimensional soil), $S^2$, to the square scaled sorptivity (unit soil), $S^{*2}$, as follows (Ross et al., 1996):

$$
S^2 = S^{*2}|h_g|K_s(\theta_s - \theta_r)
\tag{12}
$$

$S^{*2}$ can be computed by applying Eqs. (2) or (3) to the dimensionless hydraulic functions, as a function of the initial and final

water pressure heads, $h_0^*$ and $h_1^*$, or saturation degrees, $S_{e,0} = S_e(h_0^*)$ and $S_{e,1} = S_e(h_1^*)$:

$$
\begin{cases}
S_D^{2*}(S_{e,0}, S_{e,1}) = \int_{S_{e,0}}^{S_{e,1}} (S_{e,1} + S_e - 2\,S_{e,0})\,D^*(S_e)\,dS_e \\
S_K^{2*}(h_0^*, h_1^*) = \int_{h_0^*}^{h_1^*} (S_{e,1} + S_e(h^*) - 2\,S_{e,0})\,K_r(h^*)\,dh^*
\end{cases}
\tag{13}
$$

$S_D^{*2}$ and $S_K^{*2}$ define the "diffusivity" and "conductivity" forms of the squared scaled sorptivity and are related to each other by Eqs. (4) and (5), leading to:

$$
\begin{cases}
S_K^{*2}(h_0^*, h_1^* \le h_a^*) = S_D^{*2}(S_{e,0}, S_{e,1}) \\
S_K^{*2}(h_0^*, h_1^* > h_a^*) = S_D^{*2}(S_{e,0}, S_{e,1}) + 2(1 - S_{e,0})(h_1^* - h_a^*)
\end{cases}
\tag{14}
$$

With $h_a^* = 0$ for the soils with a null air-entry water pressure head or $h_a^* < 0$ otherwise. Eqs. (13) and (14) show that the dimensionless sorptivity functions, $S_D^{*2}$ and $S_K^{*2}$ depend only on the shape parameters and the initial and final conditions. Consequently, one advantage to the simplification approach is that it allows to separate the effect of different scale parameters.

To further simplify, (Haverkamp et al., 2005) isolated the contributions of the initial and final conditions to sorptivity. They considered that, at dry initial states ($\theta_0 \le \frac{1}{4}\theta_s$) and zero water pressure head at surface (i.e., $h_1 = 0$ and $\theta_1 = \theta_s$), the following

approximation applied:

$$
\begin{cases}
S^2(\theta_0, \theta_s) \approx S^2(0, \theta_s)\frac{K_s - K_0}{K_s}\frac{\theta_s - \theta_0}{\theta_s} \\
S^2(0, \theta_s) = c_p|h_g|K_s\theta_s
\end{cases}
\tag{15}
$$

Where $K_0$ corresponds to the initial hydraulic conductivity ($K_0 = K(\theta_0)$) and $c_p$ is a proportionality constant that depends only on shape hydraulic parameters (Haverkamp et al., 2005). By combining the two Eqs. (15), the sorptivity, $S^2(\theta_0, \theta_s)$, can





be defined as the product of three different terms that account for the respective contributions of the shape parameters (lumped into the hydraulic parameter $c_p$), the scale parameters $|h_g|$, $K_s$, $\theta_s$, and initial and final conditions:

$$S^2(\theta_0, \theta_s) \approx c_p |h_g| K_s \theta_s \frac{K_s - K_0}{K_s} \frac{\theta_s - \theta_0}{\theta_s} \tag{16}$$

Equation (16) offers an accurate and practical approximation for the computation of sorptivity in the case of Beerkan runs, i.e., for a zero water pressure head imposed at the soil surface, $h_1 = 0$. Equation (16) was frequently used for the treatment of Beerkan data and in particular in all BEST methods (Lassabatere et al., 2006; Yilmaz et al., 2010; Angulo-Jaramillo et al., 2019). However, it addresses the case of soils with null residual water content, $\theta_r = 0$, and without any air-entry pressure head, $h_a = 0$. Besides, it was mostly developed and used for the vGB model.

In this study, we adapt Eq. (16) to any type of hydraulic models, including those with non-null residual water contents and air-entry water pressure heads. First of all, we consider that the residual water content $\theta_r$ must be accounted for. We suggest to replace Eq. (15) with the following equation:

$$\frac{S^2(\theta_0, \theta_s)}{S^2(\theta_r, \theta_s)} \approx \frac{K_s - K_0}{K_s} \frac{\theta_s - \theta_0}{\theta_s - \theta_r} \tag{17}$$

Indeed, the denominator $\theta_s$ must be replaced by $(\theta_s - \theta_r)$ when $\theta_r \neq 0$ to ensure that the ratio $\frac{S^2(\theta_0, \theta_s)}{S^2(\theta_r, \theta_s)}$ tends towards unity when $\theta_0 \rightarrow \theta_r$. Secondly, we consider that the approximation behind Eq. (15) involves only the unsaturated part of sorptivity, i.e., $S_D^2$. As mentioned above, when the air-entry water pressure head is non-null, the computation of sorptivity $S_K^2(h_0, 0)$ must be split into its unsaturated and saturated parts, as illustrated by Eq. (5). In that case, the following derivations can be proposed:

$$
\begin{aligned}
S_K^2(h_0, h_1 = 0) &= S_D^2(\theta_0, \theta_s) + 2(\theta_s - \theta_0) K_s (h_1 - h_a) \\
&= \frac{K_s - K_0}{K_s} \frac{\theta_s - \theta_0}{\theta_s - \theta_r} S_D^2(\theta_r, \theta_s) + 2(\theta_s - \theta_0) K_s |h_a| \quad \text{since} \quad h_1 = 0 \\
&= \frac{K_s - K_0}{K_s} \frac{\theta_s - \theta_0}{\theta_s - \theta_r} S_D^2(\theta_r, \theta_s) + \frac{\theta_s - \theta_0}{\theta_s - \theta_r} \left(2(\theta_s - \theta_r) K_s |h_a|\right)
\end{aligned}
\tag{18}
$$

In the right-hand side of Eq. (18), the terms $S_D^2(\theta_r, \theta_s)$ and $2(\theta_s - \theta_r) K_s |h_a|$ refers to the unsaturated and saturated parts of the sorptivity $S_K^2(-\infty, 0)$, given that the application of Eq. (5) to the case of $h_0 = -\infty$ and $h_1 = 0$, leads to:

$$S_K^2(-\infty, 0) = S_D^2(\theta_r, \theta_s) + 2(\theta_s - \theta_r) K_s |h_a| \tag{19}$$

Consequently, Eq. (18) can be rewritten as follows:

$$S_K^2(h_0, 0) = \frac{K_s - K_0}{K_s} \frac{\theta_s - \theta_0}{\theta_s - \theta_r} \left(S_K^2(-\infty, 0) - 2(\theta_s - \theta_0) K_s |h_a|\right) + \frac{\theta_s - \theta_0}{\theta_s - \theta_r} \left(2(\theta_s - \theta_r) K_s |h_a|\right) \tag{20}$$

Then, this simplifying approach (Eq. 20), can be combined with the scaling procedure proposed by Ross et al. (1996), i.e., Eq. (11) and Eq. (12) to give the following expressions:

$$S_K^2(h_0, 0) = \left(\frac{K_s - K_0}{K_s} \frac{\theta_s - \theta_0}{\theta_s - \theta_r} \left(S_K^{2*}(-\infty, 0) - 2|h_a^*|\right) + \frac{\theta_s - \theta_0}{\theta_s - \theta_r} 2|h_a^*|\right) (\theta_s - \theta_r) K_s |h_g| \tag{21}$$





or, equivalently,

$$
\begin{cases}
S_K^2(h_0,0) = S_K^{*2}(h_0^*,0)\,(\theta_s - \theta_r)\,K_s|h_g| \\
\text{with} \\
S_K^{*2}(h_0^*,0) = (R_K R_\theta\,(c_p - 2|h_a^*|) + 2\,R_\theta|h_a^*|) \\
c_p = S_K^{2*}(-\infty,0) \\
R_\theta = \frac{\theta_s - \theta_0}{\theta_s - \theta_r} = 1 - S_{e,0} \\
R_K = \frac{K_s - K_0}{K_s} = 1 - K_r\,(S_{e,0})
\end{cases}
\tag{22}
$$

The coefficient $c_p$, as pioneered by Haverkamp et al. (2005), corresponds to the square sorptivity of the unit soil for infinite initial water pressure head and null water pressure head at surface, $h_0^* = -\infty$ and $h_1^* = 0$. $c_p$ can be quantified by using Eq. (13) or Eq. (14) with the right initial and final conditions, i.e., $(h_0^* = -\infty, h_1^* = 0)$ and $(S_{e,0} = 0, S_{e,1} = 1)$:

$$
\begin{cases}
c_p = \int_0^1 (1 + S_e)\,D^*\,(S_e)\,dS_e + 2\,|h_a^*| \\
c_p = \int_{-\infty}^0 (1 + S_e\,(h^*))\,K_r\,(h^*)\,dh^*
\end{cases}
\tag{23}
$$

Eqs. (22) is a new version of the approximation developed by Haverkamp et al. (2005), and can be applied to any type of hydraulic function. It separates the contribution of shape parameters (that lump in the term $c_p' = c_p - 2|h_a^*|$), the scale parameters, involving the product $(\theta_s - \theta_r)K_s|h_g|$, and lastly the air-entry water pressure head, in its scaled version, $|h_a^*|$. The set of Eqs. (22) makes it very easy to compute the sorptivity corresponding to zero water pressure head at surface and any initial water pressure head $h_0$ or water content $\theta_0$, provided that the values of $c_p$ are known. The following part aims at computing and tabulating $c_p$ for the hydraulic functions defined in Eqs. (6)-(10).

## 2.2   Scaling hydraulic functions

### 2.2.1   General expressions

The first steps of the determination of $c_p$ requires the computations of the dimensionless functions, $S_e(h^*)$, $K_r(S_e)$, and $D^*(S_e)$ to be injected in Eqs. (23). The application of the scaling Eqs. (11) to the hydraulic functions defined by Eqs. (6)-(10) leads to the following expressions:

$$
\begin{cases}
S_{e,d}\,(h^*) = H\,(1 + h^*) \\
K_{r,d}\,(S_e) = H\,(S_e - 1)
\end{cases}
\qquad \text{Delta model} \tag{24}
$$

$$
\begin{cases}
S_{e,BC}\,(h^*) = (1 - H\,(1 + h^*))\,|h^*|^{\lambda_{BC}} + H\,(1 + h^*) \\
K_{r,BC}\,(S_e) = S_e^{\eta_{BC}}
\end{cases}
\qquad \text{BC model} \tag{25}
$$






$$\begin{cases} S_{e,vGB}\left(h^*\right) = \left(1 + |h^*|^{n_{vGB}}\right)^{-m_{vGB}} \\ K_{r,vGB}\left(S_e\right) = S_e^{\eta_{vGB}} \end{cases} \quad \text{with} \quad m_{vGB} = 1 - \frac{2}{n_{vGB}} \qquad \text{vGB model} \qquad (26)$$

$$\begin{cases} S_{e,vGM}\left(h^*\right) = \left(1 + |h^*|^{n_{vGM}}\right)^{-m_{vGM}} \\ K_{r,vGM}\left(S_e\right) = S_e^{l_{vGM}}\left(1 - \left(1 - S_e^{\frac{1}{m_{vGM}}}\right)^{m_{vGM}}\right)^2 \end{cases} \quad \text{with} \quad m_{vGM} = 1 - \frac{1}{n_{vGM}} \qquad \text{vGM model} \qquad (27)$$

$$\begin{cases} S_{e,KG}\left(h^*\right) = erfc\left(\frac{ln(|h^*|)}{\sqrt{2}\,\sigma_{KG}}\right) \\ K_{r,KG}\left(S_e\right) = S_e^{l_{KG}}\left(\frac{1}{2}erfc\left(erfc^{-1}\left(2S_e\right) + \frac{\sigma_{KG}}{2}\right)\right)^2 \end{cases} \qquad \text{KG model} \qquad (28)$$

Where H stands for the one-sided Heaviside step function. Note that the scaling parameter for water pressure head, $h_g$, used in the Eqs. (6)-(10) was set equal to the air-entry pressure head for the Delta and BC WR functions, i.e., $h_d$ and $h_{BC}$ were set equal to $h_a$, as mentioned above.

Eqs. (24)-(28) were then used to derive the following formulations for the dimensionless diffusivity functions, $D^*\left(S_e\right)$, 200 applying $D^*\left(S_e\right) = K_r\left(S_e\right)\frac{dh^*}{dS_e}$ (see Appendix A):

$$D_d^*\left(S_e\right) = \delta\left(S_e\right) \tag{29}$$

$$D_{BC}^*\left(S_e\right)) = \frac{1}{\lambda_{BC}}S_e^{\eta_{BC} - \left(\frac{1}{\lambda_{BC}} + 1\right)} \tag{30}$$

$$D_{vGB}^*\left(S_e\right)) = \frac{1 - m_{vGB}}{2\,m_{vGB}}S_e^{\eta_{vGB} - \frac{1 + m_{vGB}}{2\,m_{vGB}}}\left(1 - S_e^{\frac{1}{m_{vGB}}}\right)^{-\frac{1 + m_{vGB}}{2}} \tag{31}$$

$$D_{vGM}^*\left(S_e\right)) = \frac{1 - m_{vGM}}{m_{vGM}}S_e^{l_{vGM} - \frac{1}{m_{vGM}}}\left(\left(1 - S_e^{\frac{1}{m_{vGM}}}\right)^{-m_{vGM}} + \left(1 - S_e^{\frac{1}{m_{vGM}}}\right)^{m_{vGM}} - 2\right) \tag{32}$$

$$D_{KG}^*\left(S_e\right)) = \frac{1}{2}\sqrt{\frac{\pi}{2}}\sigma_{KG}S_e^{l_{KG}}\left(erfc\left(erfc^{-1}\left(2\,S_e\right) + \frac{\sigma_{KG}}{\sqrt{2}}\right)\right)^2 e^{\left(erfc^{-1}\left(2\,S_e\right)\right)^2 + \sqrt{2}\,\sigma_{KG}\,erfc^{-1}\left(2\,S_e\right)} \tag{33}$$

Where $\delta$ stands for the one-sided Dirac delta function (Triadis and Broadbridge, 2012). These expressions are demonstrated in the appendix A.





### 2.2.2 Further simplifications

To reduce the number of shape parameters, we here propose several additional simplifications, that are usually considered (Angulo-Jaramillo et al., 2016). Several authors used capillary models to relate the unsaturated hydraulic conductivity to the water retention functions. For the vGB model, Haverkamp et al. (2005) linked the shape parameter related to the hydraulic conductivity, $\eta$, with the combination of those of the water retention function, $\lambda = m\,n$ as follows:

$$\eta = \frac{2}{\lambda} + 2 + p \tag{34}$$

Where the tortuosity parameter, $p$, takes the values of 1 for the case of the Burdine's condition. We also consider the same equation for the BC model given its similarity with the vGB model (as demonstrated below, in the result section). In addition, further simplifications involved the values of the tortuosity parameters, $l_{vGM}$ and $l_{KG}$ in the vGM and KG models. These were fixed at the by-default value: $l_{vGM} = l_{KG} = 1/2$ (Šimůnek et al., 2003; Kosugi, 1996; Kosugi and Hopmans, 1998). In practice, these shape parameters are rarely varied (Haverkamp et al., 2005). With those supplementary considerations, the diffusivity functions for BC and vGB models become (see demonstration in appendix A):

$$D_{BC}^{*}\left(S_e\right) = \frac{1}{\lambda_{BC}} S_e^{\frac{1}{\lambda_{BC}}+2} \tag{35}$$

$$D_{vGB}^{*}\left(S_e\right) = \frac{1-m_{vGB}}{2\,m_{vGB}} S_e^{\frac{1+3\,m_{vGB}}{2\,m_{vGB}}} \left(1 - S_e^{\frac{1}{m_{vGB}}}\right)^{-\frac{1+m_{vGB}}{2}} \tag{36}$$

$$D_{vGM}^{*}\left(S_e\right) = \frac{1-m_{vGM}}{m_{vGM}} S_e^{\frac{m_{vGM}-2}{2\,m_{vGM}}} \left(\left(1 - S_e^{\frac{1}{m_{vGM}}}\right)^{-m_{vGM}} + \left(1 - S_e^{\frac{1}{m_{vGM}}}\right)^{m_{vGM}} - 2\right) \tag{37}$$

$$D_{KG}^{*}\left(S_e\right) = \frac{1}{2}\sqrt{\frac{\pi}{2}}\sigma_{KG} S_e^{\frac{1}{2}} \left(erfc\left(erfc^{-1}\left(2\,S_e\right) + \frac{\sigma_{KG}}{\sqrt{2}}\right)\right)^2 e^{\left(erfc^{-1}\left(2\,S_e\right)\right)^2 + \sqrt{2}\,\sigma_{KG}\,erfc^{-1}\left(2\,S_e\right)} \tag{38}$$

The set of equations Eqs. (35)-(38) shows that the studied hydraulic functions and hydraulic diffusivity functions involve only one shape parameter, i.e., $\lambda_{BC}$, $m_{vGB}$, $m_{vGM}$ and $\sigma_{KG}$ for BC, vGB, vGM and KG hydraulic models, respectively. In the following, we consider the both cases of generic Eqs. (30)-(33) and simplified Eqs. (35)-(38) versions of hydraulic functions and diffusivity functions for the analytical determination of $c_p$. Then numerical applications are performed only for the simplified Eqs. (35)-(38).

### 2.3 Integral determination of parameter $c_p$

Once the dimensionless diffusivity functions were determined, the use of Eq. (23) allowed the determination of $c_p$, either analytically of numerically, depending on the considered hydraulic models.





### 2.3.1 $c_p$ for the Delta model

This case is the easiest for the determination of $c_p$. Indeed, the hydraulic conductivity function is characterized by a null hydraulic conductivity for $h^* < -1$ and a unit value $K_r = 1$ for $h^* \geq -1$, as featured by Eq. (24). The determination of $c_p$ then follows straight from Eq. (23), with the following steps:

$$
c_{p,d} = \int_{-\infty}^{0} \left(1 + S_{e,d}\left(h^*\right)\right) K_{r,d}\left(h^*\right) dh^*
$$
$$
= \int_{-\infty}^{-1} \cdot 0 \cdot dh^* + \int_{-1}^{0} \left(1+1\right) \cdot 1 \cdot dh^*
$$
$$
= 2 \tag{39}
$$

Note that this value of 2 was already proposed by (Haverkamp et al., 2005) for the "Green and Ampt" soils, as defined by these
authors.

### 2.3.2 $c_p$ for the Brooks and Corey (BC) model

The BC model involves an air-entry water pressure head $h_a^* = -1$. We used the Eq. (23) while accounting for the saturated part of sorptivity and used the diffusivity form for the determination of the unsaturated sorptivity. Indeed, the diffusivity function obeys a power law Eq. (25), which makes it possible to integrate analytically the diffusivity form of sorptivity. Simple algebraic
operations and integrations of Eq. (23) lead to the following equation (see demonstration in appendix B1):

$$
c_{p,BC}\left(\lambda_{BC}, \eta_{BC}\right) = 2 + \frac{1}{\lambda_{BC}\,\eta_{BC} - 1} + \frac{1}{\lambda_{BC}\,\eta_{BC} + \lambda_{BC} - 1} \tag{40}
$$

When the relation between $\eta_{BC}$ and $\lambda_{BC}$ is ruled by Eq. (34) with $p = 1$, the analytical expression of $c_p$ turns into:

$$
c_{p,BC}\left(\lambda_{BC}\right) = 2 + \frac{1}{3\,\lambda_{BC} + 1} + \frac{1}{4\,\lambda_{BC} + 1} \tag{41}
$$

Our results are in line with previous studies (Varado et al., 2006).

### 2.3.3 $c_p$ for the van Genuchten-Burdine (vGB) model

In this case, there is no air-entry pressure head, i.e., $h_a^* = 0$. Eq. (23) shows that $c_p$ reverts to the diffusivity form of the squared dimensionless sorptivity, $\int_0^1 \left(1 + S_e\right) D^*(S_e)dS_e$. Besides, the diffusivity function Eq. (26) makes the dimensionless sorptivity analytically integrable, leading to (see demonstration in appendix B2):

$$
c_{p,vGB}\left(m_{vGB}, n_{vGB}, \eta_{vGB}\right) = \Gamma\left(1 + \frac{1}{n_{vGB}}\right) \left[\frac{\Gamma\left(m_{vGB}\,\eta_{vGB} - \frac{1}{n_{vGB}}\right)}{\Gamma\left(m_{vGB}\,\eta_{vGB}\right)} + \frac{\Gamma\left(m_{vGB}\,\eta_{vGB} + m_{vGB} - \frac{1}{n_{vGB}}\right)}{\Gamma\left(m_{vGB}\,\eta_{vGB} + m_{vGB}\right)}\right] \tag{42}
$$

where $\Gamma$ is the gamma function:

$$
\Gamma(z) = \int_{0}^{+\infty} t^{z-1}\,e^{-t}\,dt \tag{43}
$$





Considering the relations between $m$ and $n$ Eq. (34) and the relation between $\eta$ and $\lambda = m\,n$ Eq. (34), the following simplification comes out:

$$c_{p,vGB}\left(m_{vGB}\right) = \Gamma\left(\frac{3-m_{vGB}}{2}\right)\left[\frac{\Gamma\left(\frac{1+5\,m_{vGB}}{2}\right)}{\Gamma\left(1+2\,m_{vGB}\right)} + \frac{\Gamma\left(\frac{1+7\,m_{vGB}}{2}\right)}{\Gamma\left(1+3\,m_{vGB}\right)}\right] \tag{44}$$

The expression corresponding to Eq. (42) was already proposed and discussed by (Haverkamp et al., 2005).

### 2.3.4 $c_p$ for the van Genuchten-Mualem (vGM) model

In contrast with the vGB model, no analytical expressions were reported in the literature for this model. By analogy with the case of vGB functions, analytical developments were proposed to analytically integrate the diffusivity form of the dimensionless sorptivity, leading to the following analytical expression for $c_p$ (see demonstration in appendix B3):

$$c_{p,vGM}\left(m_{vGM},l_{vGM}\right) = \Gamma(2-m_{vGM})\left(\frac{\Gamma(m_{vGM}(1+l_{vGM}))}{(m_{vGM}(1+l_{vGM})-1)\,\Gamma(m_{vGM}\,l_{vGM})} + \frac{\Gamma(m_{vGM}(2+l_{vGM}))}{(m_{vGM}(2+l_{vGM})-1)\,\Gamma(m_{vGM}(1+l_{vGM}))}\right)$$
$$+ (1-m_{vGM})\left[\left(\frac{\Gamma(m_{vGM}(1+l_{vGM}))\,\Gamma(1+m_{vGM})}{(m_{vGM}(1+l_{vGM})-1)\,\Gamma(m_{vGM}(2+l_{vGM}))} + \frac{\Gamma(m_{vGM}(2+l_{vGM}))\,\Gamma(1+m_{vGM})}{(m_{vGM}(2+l_{vGM})-1)\,\Gamma(m_{vGM}(3+l_{vGM}))}\right)\right.$$
$$\left. - 2\left(\frac{1}{m_{vGM}(1+l_{vGM})-1} + \frac{1}{m_{vGM}(2+l_{vGM})-1}\right)\right] \tag{45}$$

Note that, this equation requires that $m_{vGM} \neq \frac{1}{1+l_{vGM}}$ and $m_{vGM} \neq \frac{1}{2+l_{vGM}}$. Considering that shape parameter $l_{vGM} = \frac{1}{2}$, as usually considered, Eq. (45) can be simplified to:

$$c_{p,vGM}\left(m_{vGM}\right) = \Gamma(2-m_{vGM})\left(\frac{\Gamma\left(\frac{3m_{vGM}}{2}\right)}{\left(\frac{3m_{vGM}}{2}-1\right)\Gamma\left(\frac{m_{vGM}}{2}\right)} + \frac{\Gamma\left(\frac{5m_{vGM}}{2}\right)}{\left(\frac{5m_{vGM}}{2}-1\right)\Gamma\left(\frac{3m_{vGM}}{2}\right)}\right)$$
$$+ (1-m_{vGM})\left[\left(\frac{\Gamma\left(\frac{3m_{vGM}}{2}\right)\Gamma(1+m_{vGM})}{\left(\frac{3m_{vGM}}{2}-1\right)\Gamma\left(\frac{5m_{vGM}}{2}\right)} + \frac{\Gamma\left(\frac{5m_{vGM}}{2}\right)\Gamma(1+m_{vGM})}{\left(\frac{5m_{vGM}}{2}-1\right)\Gamma\left(\frac{7m_{vGM}}{2}\right)}\right) - 2\left(\frac{1}{\frac{3m_{vGM}}{2}-1} + \frac{1}{\frac{5m_{vGM}}{2}-1}\right)\right] \tag{46}$$

These sets of equations have never been proposed and constitute one of the inputs of this study. The complexity of algebraic
developments for the derivation of Eqs. (42) and (44) make them valuable.

### 2.3.5 $c_p$ for the Kosugi (KG) model

No analytical formulation was found for the case of Kosugi's hydraulic functions. Therefore, the dimensionless scaled sorptivity was computed numerically with a generic procedure that can be applied to any type of model for WRHC functions. To avoid integration over infinite intervals with respect to h and integration of an infinite diffusivity close to saturation, the integral was





split into two parts, leading to the following developments:

$$
\begin{aligned}
c_{p,KG}\left(\sigma_{KG}, l_{KG}\right) &= \int_{-\infty}^{0}\left(1+S_{e,KG}\left(h^{*}\right)\right)K_{r,KG}\left(h^{*}\right)dh^{*}\\
&= \int_{-\infty}^{h^{*}_{KG}\left(\frac{1}{2}\right)}\left(1+S_{e,KG}\left(h^{*}\right)\right)K_{r,KG}\left(h^{*}\right)dh^{*}+\int_{h^{*}_{KG}\left(\frac{1}{2}\right)}^{0}\left(1+S_{e,KG}\left(h^{*}\right)\right)K_{r,KG}\left(h^{*}\right)dh^{*}\\
&= \int_{0}^{\frac{1}{2}}\left(1+S_{e}\right)D^{*}_{KG}\left(S_{e}\right)dS_{e}+\int_{h^{*}_{KG}\left(\frac{1}{2}\right)}^{0}\left(1+S_{e,KG}\left(h^{*}\right)\right)K_{r,KG}\left(h^{*}\right)dh^{*}
\end{aligned}
\tag{47}
$$

where $h^{*}_{KG}\left(\frac{1}{2}\right)$ is the water pressure head corresponding to $S_{e}=\frac{1}{2}$. In the last version of Eq. (47), $c_{p}$ is composed of two integrals of continuous functions over closed intervals, and thus integrable at all times. Again, a simplified version is proposed assuming that the shape parameter $l_{KG}$ is fixed at: $l_{KG}=\frac{1}{2}$.

## 285 2.4 Shape indexes for comparing $c_p$ between the selected hydraulic models

The approach described below allows $c_{p}$ determination for the selected BC, vGB, vGM, and KG models. The dependency of $c_{p}$ upon the chosen hydraulic model could be questioned. Indeed, in addition to its contribution to simplifying Eq. (22), $c_{p}$ has real physical meaning: it corresponds to the sorptivity of unit soils for the case of zero water pressure head at the soil surface and utterly dry initial profile. Therefore the dependency of such sorptivity upon the selected hydraulic model has also 290 to be questioned. Consequently, we designed shape indexes that allow comparing $c_{p}$ between models. These shape indexes were built to describe the same state of the WR functions regardless of the chosen hydraulic model. We designed these shape indexes to vary WR function between two extreme states: (i) a value close to zero for gradual WR function corresponding to soils with broad pore size distributions, (ii) a value of unity for stepwise WR function mimicking an abrupt change of water content corresponding to soils with very narrow pore size distributions. This section presents the design of these shape indexes.

A sensitivity analysis of vGB and vGM models showed that the parameter $m$ is adequate for varying the WR functions between a gradual shape $(m=0)$ and a stepwise function for $(m=1)$. We thus consider $m_{vGB}$ and $m_{vGM}$ to be the appropriate shape indexes, i.e., $x_{vGB}=m_{vGB}$ and $x_{vGM}=m_{vGM}$. Next, we designed the shape index for BC, $x_{BC}$, deriving from vGB, $m_{vGB}$, considering that vGB and BC functions describe close WR curves. Indeed, for large values of $n_{vGB}$, the vGB model converge towards a power function similar to the BC model (Haverkamp et al., 2005):

$$
\lim_{n_{vGB}\to+\infty}\left(1+\left|h^{*}\right|^{n_{vGB}}\right)^{-m_{vGB}}\approx\left|h^{*}\right|^{-n_{vGB}\,m_{vGB}}
$$

$$
\approx\left|h^{*}\right|^{-\lambda_{vGB}}\quad\text{with}\quad\lambda_{vGB}=n_{vGB}\,m_{vGB}
\tag{48}
$$

The equation $\lambda_{BC}=n_{vGB}-2$ defines a relation between $\lambda_{BC}$ and then $n_{vGB}$ that ensures a similar state for WR functions. Substituting $n_{vGB}$ by $m_{vGB}$ according to $m_{vGB}=1-\frac{2}{n_{vGB}}$ leads to:

$$
m_{vGB}=\frac{\lambda_{BC}}{2+\lambda_{BC}}
\tag{49}
$$



Since $m_{vGB}$ is the appropriate shape index for the vGB model, we consider its equivalent, $\frac{\lambda_{BC}}{2+\lambda_{BC}}$, as the appropriate shape

index for the BC model, leading to:

$$x_{BC} = \frac{\lambda_{BC}}{2+\lambda_{BC}} \tag{50}$$

For KG functions, we considered that stepwise WR functions are associated with a narrow pore size distribution, i.e., a null

standard deviation, $\sigma_{KG}$. In contrast, gradual WR functions correspond to a spread distribution of pore size distribution, i.e.

very large values of $\sigma_{KG}$. Consequently, by analogy with Eq. (50), i.e. by using a ratio, we propose the following shape index,

$\sigma_{KG}$:

$$x_{KG} = \frac{1}{1+\sigma_{KG}} \tag{51}$$

Finally, the shape parameters for each model can be expressed as a function of the shape index, by inverting the previous

equations. For the sake of simplicity, we use the same letter "$x$", to denote the different shape indexes, $x_{BC}$, $x_{vGB}$, $x_{vGM}$, or

$x_{KG}$. We obtain the following relations:


$$\begin{cases} \lambda_{KG} = \frac{2x}{1-x} \\ m_{vGB} = x \\ m_{vGM} = x \\ \sigma_{KG} = \frac{1-x}{x} \end{cases} \tag{52}$$

Where $x$ takes values in the interval $[0,1]$. Equations (52) provide the shape parameters of the studied models for a given value

of shape index $x$, i.e., for a similar state of WR function between gradual ($x=0$) and stepwise functions ($x=1$).

On the basis on these relations between shape parameters and indexes, $c_p$ can easily be related to the shape index using

Eq. (41), Eq. (44) and Eq. (46) (obtained for the simplified diffusivity functions, Eqs. (35)-(38). For the KG model, the com-

putations of $c_p$, remained numerical. The following set of equations were obtained:

$$\begin{cases} c_{p,BC}(x) = 2 + \frac{1-x}{5x+1} + \frac{1-x}{7x+1} \\ c_{p,vGB}(x) = \Gamma\left(\frac{3-x}{2}\right)\left[\frac{\Gamma\left(\frac{1+5x}{2}\right)}{\Gamma(1+2x)} + \frac{\Gamma\left(\frac{1+7x}{2}\right)}{\Gamma(1+3x)}\right] \\ c_{p,vGM}(x) = \Gamma(2-x)\left[\frac{\Gamma\left(\frac{3}{2}x\right)}{\left(\frac{3}{2}x-1\right)\Gamma\left(\frac{1}{2}x\right)} + \frac{\Gamma\left(\frac{5}{2}x\right)}{\left(\frac{5}{2}x-1\right)\Gamma\left(\frac{3}{2}x\right)}\right] + (x-1)\left[\frac{\Gamma\left(\frac{3}{2}x\right)\Gamma(1+x)}{\left(\frac{3}{2}x-1\right)\Gamma\left(\frac{5}{2}x\right)} + \frac{\Gamma\left(\frac{5}{2}x\right)\Gamma(1+x)}{\left(\frac{5}{2}x-1\right)\Gamma\left(\frac{7}{2}x\right)} - 2\left(\frac{1}{\frac{3}{2}x-1} + \frac{1}{\frac{5}{2}x-1}\right)\right] \\ c_{p,KG}(x) = \int_0^{\frac{1}{2}} (1+S_e) D_{KG(x)}(S_e) dS_e + \int_{h_{KG(x)}^*\left(\frac{1}{2}\right)}^0 \left(1+S_{e,KG(x)}(h^*)\right) K_{r,KG}(h^*) dh^* \quad \text{with} \quad \sigma_{KG}(x) = \frac{1-x}{x} \end{cases} \tag{53}$$

We performed a sensitivity analysis by varying the shape index $x$ for each model between 0 and 1 by increments of 0.025. We

then computed the different shape parameters for BC, vGB, vGM and KG models using Eq. (52) and then plotted the related

hydraulic functions Eqs. (25)-(28), with $\eta = \frac{2}{\lambda} + 2 + p$ and $l_{vGM} = l_{KG} = \frac{1}{2}$) and diffusivity functions Eqs. (35)-(38). Lastly,





we computed the squared dimensionless scaled sorptivity $c_p$ as a function of the shape index, Eqs. (53) and discussed the function $c_p(x)$ regarding the choice of the hydraulic function. The values of $c_p(x)$ are also compared to that of Delta model (Dirac delta functions), i.e. $c_{p,d} = 2$.

## 3   Results

### 3.1   Analysis of the hydraulic functions and diffusivity

The hydraulic functions and diffusivity functions are plotted in Fig. 1 for the shape index value of 0.275, and their sensitivity upon each model's shape index is shown in Fig. 2. For the sake of clarity, we plotted the relative hydraulic conductivity both as functions of saturation degree, $K_r(S_e)$, and water pressure head, $K_r(h^*)$, noting that these functions have distinct uses: $K_r(S_e)$ defines the HC functions as a property of the soils, whereas $K_r(h^*)$ is mostly used to compute sorptivity, e.g., as in Eq. (13). Thus, $K_r(h^*)$ has a similar role as the diffusivity function $D^*(S_e)$, and the shapes and properties of these two

functions determine the values of the squared scaled sorptivity $c_p$.

The comparison between the hydraulic models (at the same shape index value) reveals some similarities and discrepancies (Fig. 1). Three of the water retention models (vGB, vGM, KG) exhibit an inflection point with a continuous increase in $S_e(h^*)$ over the whole interval $(-\infty, 0]$ (Fig. 1a, "vGB", "vGM", and "KG"), while the BC model reaches the asymptote $S_e = 1$ at $h^* = -1$ with full saturation for $h^* \geq -1$ (Fig. 1a, "BC"). Despite that difference, the BC and vGB models exhibit

similar shapes (Fig. 1a, "BC" versus "vGB"). The vGM model exhibits a more progressive increase in $S_e(h^*)$ while remaining asymmetrically distributed across the inflection point (Fig. 1a, "vGM"). Lastly, the KG model exhibits an even more progressive increase and a perfect symmetry around the inflection point (Fig. 1a, "KG"). The position of the inflection points depends on the chosen hydraulic model. By construction, the inflection point is positioned at $h^* = -1$ for the KG model. The others models have inflection points positioned at larger abscises (in absolute values), with similar intermediate values for the BC and the

vGB models and the largest abscises for the vGM model (Fig. 1a).

Regarding the relative hydraulic conductivity, the BC and vGB models have similar shapes for $K_r(S_e)$, with both typical of power functions (Fig. 1b, "BC and "vGB"). In contrast, the vGM and KG models depict an inflection point, with larger increase both at low saturation degrees and close to saturation compared to intermediate saturation degrees. In particular, these two models exhibit a very large increase close to saturation whereas BC and vGB models have a gradual increase (Fig. 1b,

"vGM" and "KG" versus "BC" and "vGB", close to $S_e = 1$). This feature allows the vGM and KG models to simulate large drops in hydraulic conductivity close to saturation that are typical for certain soils. The functions $K_r(h^*)$ are depicted in Fig. 1c and combine the properties of the functions $K_r(S_e)$ and $S_e(h^*)$, as described depicted above. The function $K_r(h^*)$ exhibit similar shapes for BC and vGB models, with a quasi linear and sharp increase for $h^* \leq -1$ followed by a plateau (Fig. 1c, "BC" and "vGB"). The vGM and KG models exhibit a much more progressive increase, involving a much larger

range of water pressure heads. This feature is the most pronounced for the KG model. This more progressive increase reflects the more gradual WR functions, $S_e(h^*)$, as described in Fig. 1a combined with the drop in $K_r(S_e)$ that reaches unity only for saturation degrees extremely close to unity (Fig. 1b).



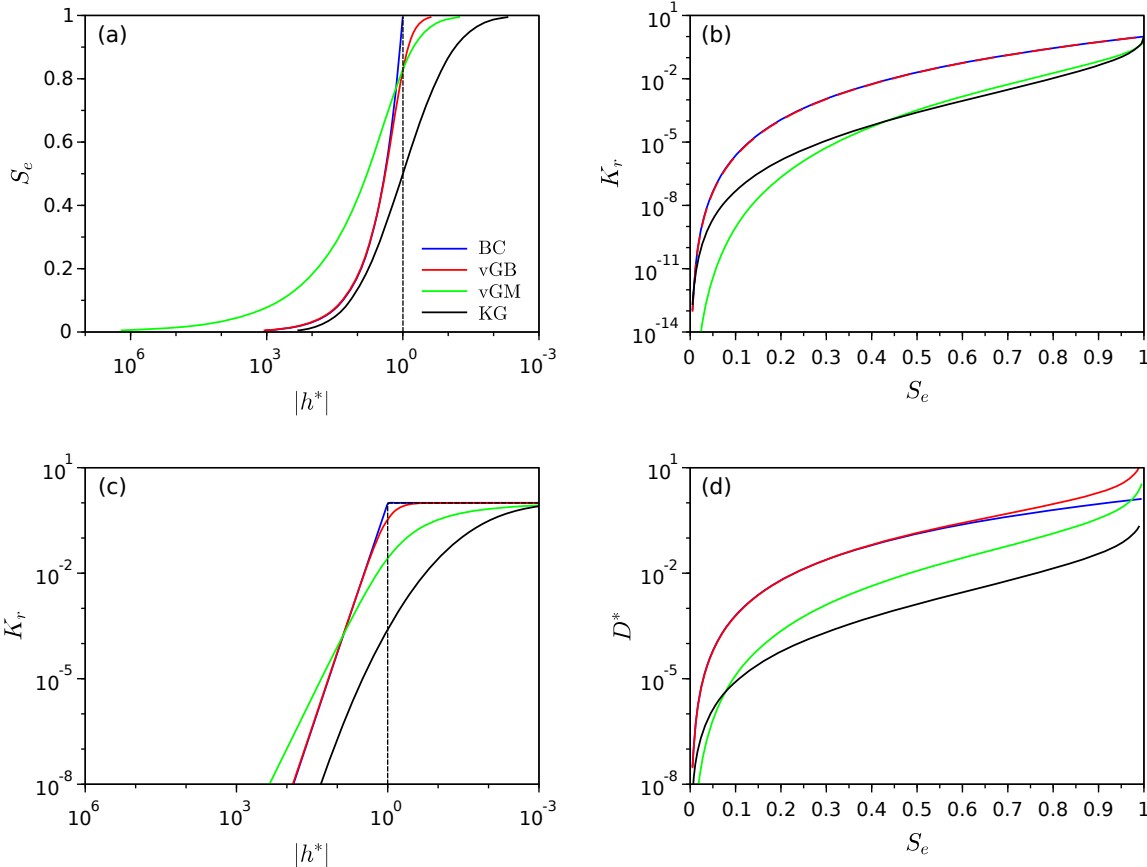

**Figure 1.** Examples of water retention, $S_e(h^*)$ (a), relative unsaturated hydraulic, $K_r(S_e)$ (b) and $K_r(h^*)$ (c), and diffusivity $D^*(S_e)$ functions (d) for the four hydraulic models: Brooks and Corey (BC), van Genuchten – Burdine (vGB), van Genuchten-Mualem (vGM), and Kosugi (KG); the curves were plotted for a value of the shape index $x$ of 0.275. The hydraulic parameters $\lambda_{BC}$, $m_{vGM}$, $m_{vGB}$, and $\sigma_{KG}$ were computed as a function of $x$ using Eq. (52) with $l_{vGM} = l_{KG} = \frac{1}{2}$. The dashed line represents the "delta" model.

As for the WR and HC functions, the diffusivity functions exhibit close shapes for the BC and vGB models (Fig. 1d). The BC model defines a concave shape with a finite maximum equal to $\lambda_{BC}$ obtained at $S_e = 1$, in line with the use of Eq. (35) at

$S_e = 1$ (Fig. 1d, "BC"). In opposite, the vGB model diverts from the concave shape to tend towards infinity close to saturation, at $S_e = 1$ (Fig. 1d, "vGB" versus "BC"). The vGB model defines a S shape that reflects the larger increases at both low and high saturation degrees with lower increase at intermediate saturation degrees (Fig. 1d, "vGB"). The two other models, vGM and KG, exhibit the same type of S shape with an infinite limit close to saturation (Fig. 1d, "vGM" and "KG"). Such infinite limit spoils the numerical integration of Eqs. (23) for the determination of $c_p$, requiring the use of the mixed formulation for

the KG models defined by Eqs. (47) and Eqs. (53).





Varying the shape index changed the WRHC functions in an expected way (Fig. 2). For the WR functions, increasing the shape index from 0 to 1 makes the shift from a gradual and moderate to an abrupt increase in saturation degree, respectively. Values close to unity makes the WR functions close to a stepwise function corresponding to the Delta model (Fig. 2, $1^{st}$ column, arrows). As for the WR functions, the increase in the shape index put the curves $K_r(h^*)$ close to stepwise functions (Fig. 2,

$3^{rd}$ column, arrows). For the BC model, we notice a decrease in $K_r(h^*)$ for $h^* \leq -1$ whereas $K_r(h^*)$ remains equal to unity above (Fig. 2c, 3rd column). In opposite, for the vGB, vGM and KG models, the increase in the shape index has two antagonist effects: a decrease of $K_r(h^*)$ for $h^* \leq -1$ followed by an increase for $h^* \geq -1$ (Fig. 2g,k,n, $3^{rd}$ column, arrows). Briefly, as expected, the water retention and the relative hydraulic conductivity tend towards stepwise functions when the shape index tends towards unity (Fig. 2, $1^{st}$ and $3^{rd}$ columns). This trend is less evident for the diffusivity functions (Fig. 2, $4^{th}$ column).

These results show that, regardless of the selected model, increasing the shape index put the hydraulic functions closer to the Delta models that correspond to soils with narrow pore size distribution. In opposite, very small values of the shape index ensure very gradual shapes for WR and HC functions. However, the results point at contrasting trends when the shape index is decreased towards zero. It is clear that for the vGB and BC models, the relative hydraulic conductivity $K_r(h^*)$ is not greatly impacted close to $h^* = 0$ (Fig. 2c,g). In opposite, for the vGM and KG models, $K_r(h^*)$ tends towards zero in the vicinity of

$h^* = 0$ (Fig. 2k,o, inverted arrows). Similarly, the dimensionless diffusivity $D^*(S_e)$ tends towards zero over the whole interval $[0, 1]$ when the shape index tends towards zero (Fig. 2l,p, inverted arrows). Consequently, the features of $K_r(h^*)$ and $D^*(S_e)$ functions presume very small values of the square scaled sorptivity for vGM and KG models, when the shape index tends towards zero (see results below).

### 3.2 Squared dimensionless scaled sorptivity as a function of shape indexes

The squared dimensionless scaled sorptivity $c_p(x)$ is plotted as its function to the shape index, $x$ (Fig. 3). The results show contrasting evolutions. For the BC model, we note a decrease of $c_{p,BC}(x)$ from 4 down to 2 (Fig. 3). Such a decrease is expected since $K_{r,BC}(h^*)$ functions decrease over the whole interval $(-\infty, 0]$ with the shape index (see Fig. 1c). This leads to the decrease of the integral involved in Eq. (23), and thus $c_p$. The upper and lower limits can be easily determined by applying Eq. (53), leading to $c_{p,BC}(0) = 4$ and $c_{p,BC}(1) = 2$.

In contrary to the BC model, $c_{p,vGB}(x)$ for the vGB model does not decrease monotonically (Fig. 3c). Instead, $c_{p,vG}(x)$ decreases to 1.5 before increasing up to 2 for $x \geq 0.52$. This feature is line with the effect of the shape index on the relative hydraulic conductivity, $K_r(h^*)$, as described above. The shape index has two antagonist effects: a decrease of $K_r(h^*)$ for $h^* \leq -1$ and an increase for $h^* \geq -1$ (Fig. 2g, arrows). The numerical computation sorts out these contrasting effects and demonstrates the two-step variation, i.e., a decrease followed by an increase. The two boundaries of the function $c_{p,vG}(x)$

can be easily found by using Eq. (53) with a lower limit of $c_{p,vGB}(0) = 2\Gamma\left(\frac{3}{2}\right)\Gamma\left(\frac{1}{2}\right) = \Gamma\left(\frac{1}{2}\right)^2 = \pi$ and an upper limit of $c_{p,vGB}(1) = \Gamma(1)\left[\frac{\Gamma(3)}{\Gamma(3)} + \frac{\Gamma(4)}{\Gamma(4)}\right] = 2$.

For the KG and vGM models, the trend is opposite and the functions $c_{p,KG}(x)$ and $c_{p,vGM}(x)$ both increase (Fig 3b,d). This increase is in line with the fact that the shape indexes mainly increase the diffusivity function, $D^*(S_e)$ over the whole intervals $[0, 1]$ and thus the integral of Eq. (23). The two functions $c_{p,KG}(x)$ and $c_{p,vGM}(x)$ increase from zero up to 2. For the

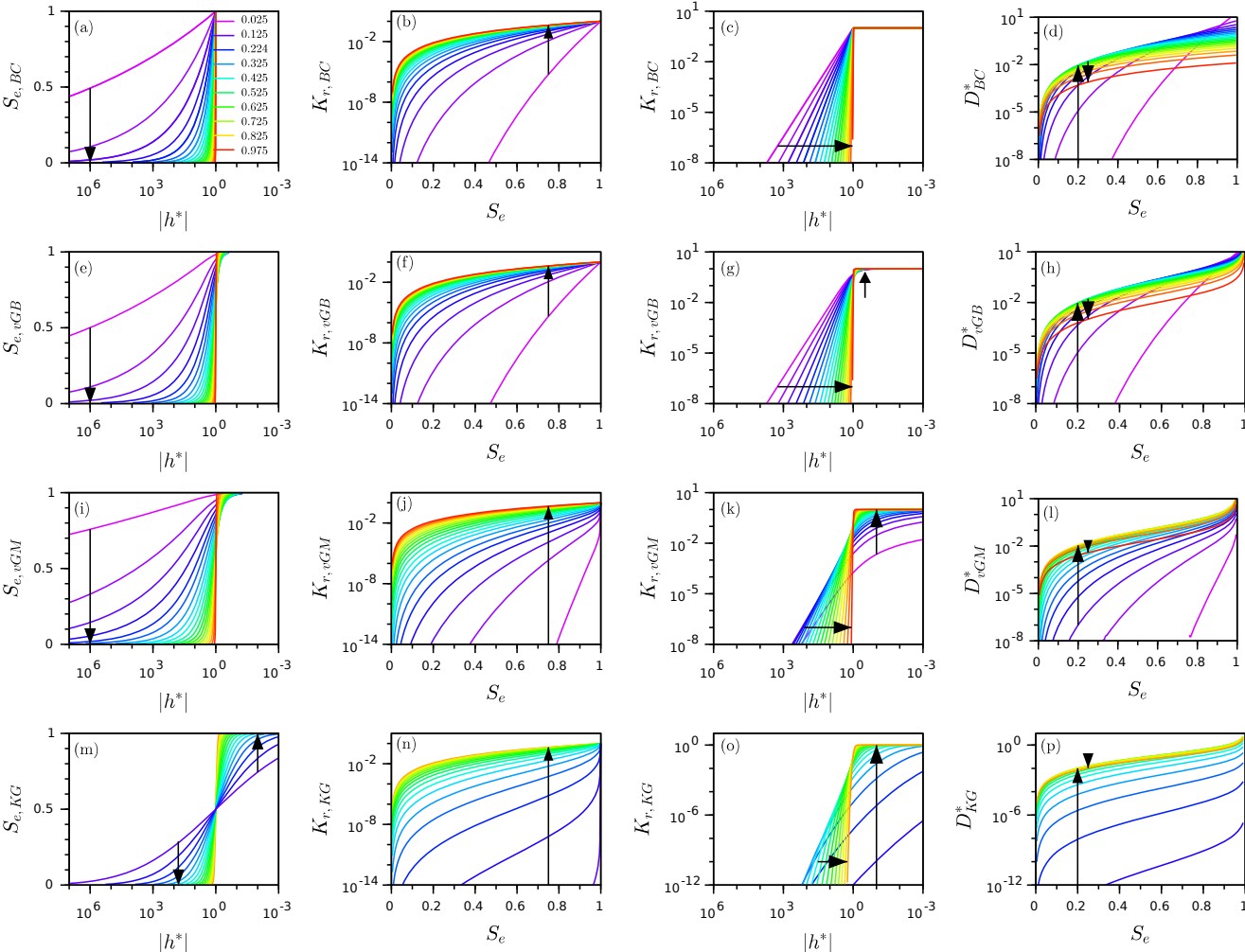

**Figure 2.** : Impact of the shape index, $x$, on the WRHC functions versus the selected hydraulic models: WR functions, $S_e(h^*)$ ($1^{st}$ column), HC functions, $K_r(S_e)$ ($2^{nd}$ column) and $K_r(h^*)$ ($3^{rd}$ column), and diffusivity function $D^*(S_e)$ functions; Brooks and Corey (BC) ($1^{st}$ row), van Genuchten – Burdine (vGB) ($2^{nd}$ row), van Genuchten-Mualem (vGM) ($3^{rd}$ row), and Kosugi (KG) models ($4^{rd}$ row); the arrows indicate increasing values of the shape index x. The hydraulic parameters $\lambda_{BC}$, $m_{vGM}$, $m_{vGB}$, and $\sigma_{KG}$ were computed as a function of $x$ using Eq. (52) with $l_{vGM} = l_{KG} = \frac{1}{2}$.

vGM model, the lower and upper limits can be demonstrated using Eq. (53) leading to $c_{p,vGM}(0) = 0$ (see appendix A4) and $c_{p,vGM}(1) = \Gamma(1)\left[\frac{\Gamma\left(\frac{1}{2}\right)}{\Gamma\left(\frac{1}{2}\right)} + \frac{\Gamma\left(\frac{3}{2}\right)}{\Gamma\left(\frac{3}{2}\right)}\right] = 2$. For KG hydraulic functions, the lower and upper limits were determined numerically, leading also to 0 and 2, with null values over a large interval of shape index, i.e., $[0, 0.3]$ (Fig. 3b).

    The four functions $c_{p,BC}(x)$, $c_{p,vGB}(x)$, $c_{p,vGM}(x)$, and $c_{p,KG}(x)$ all reach the value of 2 when the shape index approaches unity, i.e., when the WR and HC functions tend towards stepwise functions. In fact, the value of $c_p(x)$ converges to the value



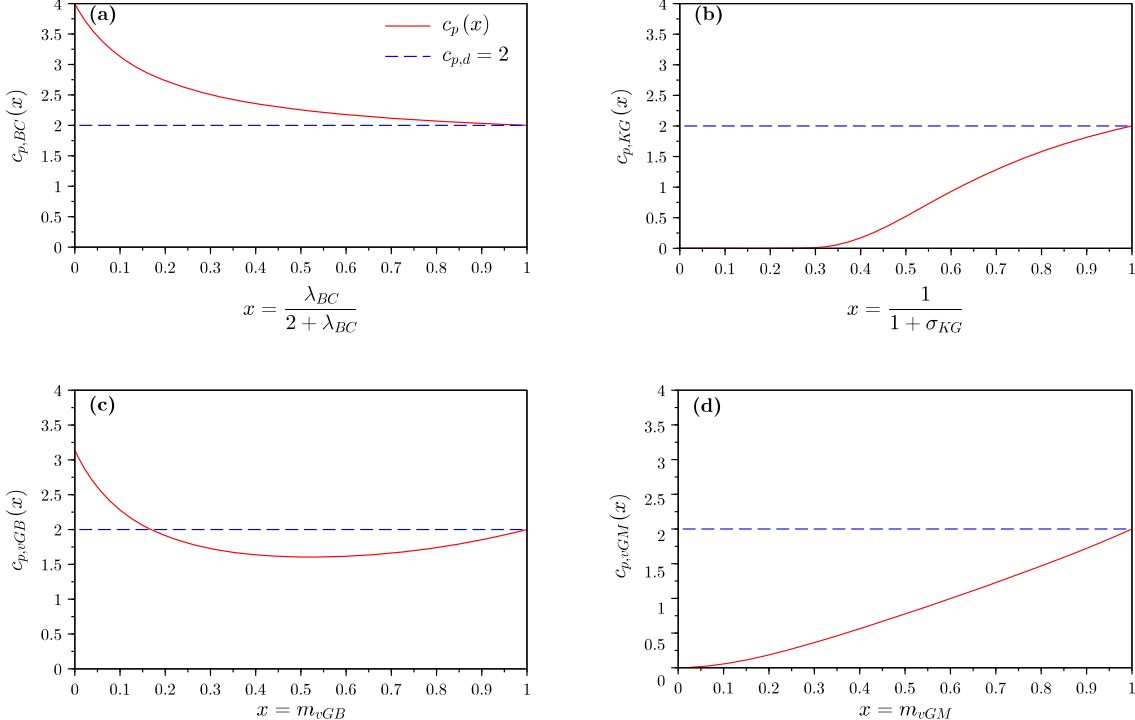

**Figure 3.** : Squared scaled sorptivity, $c_p$, as a function of shape index, $x$, for the four hydraulic models: (a) Brooks and Corey (BC), (b) Kosugi (KG), (c) van Genuchten – Burdine (vGB), and (d) van Genuchten-Mualem (vGM); the models were sorted to have on the left the broadly decreasing functions and on the right the increasing functions.

obtained for the Delta model, $c_{p,d} = 2$ (see Eq. 39). Such a result indicates that similar results should be obtained for soils with a narrow pore size distribution (like coarse soils with narrow pore size distributions), regardless of the selected model for describing their WRHC functions. In other words, the choice of the hydraulic model should not matter for soils with narrow pore size distributions. In opposite, contrasting trends are obtained when the shape index tends towards zero: quasi-null values for the vGM and KG models, $\pi$ for the vGB model and 2 for the BC model. The null values of $c_p$ for the KG and vGB models

are in line with the large decrease in $D^*(S_e)$ over the whole interval $[0,1]$ with the shape index, $x$ (see Fig. 2l,p and comments above). Such results show that the choice of the hydraulic model matters for soils with graded pore size distributions. Between these two extreme states, the four functions $c_{p,BC}(x)$, $c_{p,vGB}(x)$, $c_{p,vGM}(x)$, and $c_{p,KG}(x)$ exhibit contrasting evolutions, with $c_{p,vGM}(x)$, and $c_{p,KG}(x)$ increasing monotonously, $c_{p,BC}(x)$ decreasing monotonously, $c_{p,vGB}(x)$ exhibiting a two-step behavior of decreasing followed by increasing values.

These contrasting evolutions of functions $c_{p,BC}(x)$, $c_{p,vGB}(x)$, $c_{p,vGM}(x)$, and $c_{p,KG}(x)$ point at different implications with regards the physics of flow and infiltration in soils. It should be borne in mind that the choice of the hydraulic models should not impact the value of sorptivity, $S_K^2(h_0, 0)$ for a given soil and given initial conditions ($h_0$). Indeed, sorptivity should





**Table 1.** Values of the square dimensionless scaled sorptivity $c_p$ as a function of the shape index $x$ for the studied hydraulic functions Brooks and Corey (BC), van Genuchten – Burdine (vGB), van Genuchten-Mualem (vGM), and Kosugi (KG) functions.

| $x$ | BC | vGB | vGM | KG | $x$ | BC | vGB | vGM | KG |
|---|---|---|---|---|---|---|---|---|---|
| 0.00 | 4.000 | 3.142 ($\pi$) | 0.000 | 0.000 | ... | ... | ... | ... | .. |
| 0.02 | 3.749 | 2.891 | 2.559 E-03 | 3.38 E-776 | 0.52 | 2.237 | 1.605 | 0.820 | 0.604 |
| 0.04 | 3.549 | 2.692 | 9.814 E-03 | 2.14 E-188 | 0.54 | 2.220 | 1.605 | 0.864 | 0.686 |
| 0.06 | 3.384 | 2.529 | 2.126 E-02 | 3.106 E-81 | 0.56 | 2.205 | 1.608 | 0.908 | 0.768 |
| 0.08 | 3.246 | 2.393 | 3.646 E-02 | 2.125 E-44 | 0.58 | 2.191 | 1.611 | 0.953 | 0.848 |
| 0.10 | 3.129 | 2.279 | 5.500 E-02 | 1.161 E-27 | 0.60 | 2.177 | 1.616 | 0.998 | 0.927 |
| 0.12 | 3.028 | 2.183 | 7.654 E-02 | 9.772 E-19 | 0.62 | 2.164 | 1.623 | 1.043 | 1.004 |
| 0.14 | 2.940 | 2.099 | 0.101 | 1.875 E-13 | 0.64 | 2.151 | 1.631 | 1.089 | 1.079 |
| 0.16 | 2.863 | 2.028 | 0.127 | 4.350 E-10 | 0.66 | 2.140 | 1.640 | 1.135 | 1.151 |
| 0.18 | 2.794 | 1.966 | 0.156 | 7.998 E-08 | 0.68 | 2.128 | 1.651 | 1.181 | 1.220 |
| 0.20 | 2.733 | 1.911 | 0.187 | 3.107 E-06 | 0.70 | 2.117 | 1.662 | 1.228 | 1.287 |
| 0.22 | 2.678 | 1.864 | 0.219 | 4.429 E-05 | 0.72 | 2.107 | 1.676 | 1.275 | 1.351 |
| 0.24 | 2.629 | 1.823 | 0.253 | 3.218 E-04 | 0.74 | 2.097 | 1.690 | 1.322 | 1.412 |
| 0.26 | 2.584 | 1.787 | 0.288 | 1.463 E-03 | 0.76 | 2.088 | 1.706 | 1.370 | 1.471 |
| 0.28 | 2.543 | 1.755 | 0.325 | 4.760 E-03 | 0.78 | 2.079 | 1.723 | 1.418 | 1.527 |
| 0.30 | 2.506 | 1.727 | 0.362 | 1.212 E-02 | 0.80 | 2.070 | 1.741 | 1.467 | 1.580 |
| 0.32 | 2.471 | 1.704 | 0.401 | 2.568 E-02 | 0.82 | 2.062 | 1.761 | 1.516 | 1.630 |
| 0.34 | 2.440 | 1.683 | 0.440 | 4.735 E-02 | 0.84 | 2.054 | 1.782 | 1.566 | 1.680 |
| 0.36 | 2.410 | 1.665 | 0.480 | 7.837 E-02 | 0.86 | 2.046 | 1.804 | 1.617 | 1.727 |
| 0.38 | 2.383 | 1.650 | 0.521 | 0.119 | 0.88 | 2.039 | 1.828 | 1.669 | 1.771 |
| 0.40 | 2.358 | 1.637 | 0.562 | 0.170 | 0.90 | 2.032 | 1.853 | 1.721 | 1.814 |
| 0.42 | 2.334 | 1.627 | 0.604 | 0.229 | 0.92 | 2.025 | 1.879 | 1.775 | 1.854 |
| 0.44 | 2.312 | 1.619 | 0.646 | 0.295 | 0.94 | 2.018 | 1.907 | 1.829 | 1.893 |
| 0.46 | 2.291 | 1.613 | 0.689 | 0.367 | 0.96 | 2.012 | 1.937 | 1.885 | 1.930 |
| 0.48 | 2.272 | 1.608 | 0.732 | 0.444 | 0.98 | 2.006 | 1.967 | 1.942 | 1.966 |
| 0.50 | 2.254 | 1.606 | 0.776 | 0.523 | 1.00 | 2.000 | 2.000 | 2.000 | 2.000 |

remain constant regardless of the choice of hydraulic models since it always equals the ratio between the cumulative infiltration and the square root of time for gravity-free infiltration, as illustrated by Eq. (1). However, the choice of the hydraulic model

strongly impacts the estimation of $c_p$, for soils with broad pore size distributions, i.e., when the shape index gets close to zero: the BC and vGB models predicts non-null values for $c_p$, thus ensuring non-null values of the sorptivity (see Eq. (20)), whereas the KG or vGM models predict quasi-null values of $c_p$. We then expect the product of scale parameters "$(\theta_s - \theta_r) K_s |h_g|$" to compensate for the very low values of $c_p$ (see Eq. (20)). Given that the scale parameters $\theta_s$, $\theta_r$, and $K_s$, characterize dry



(residual) or saturated states of the soil, these parameters are not expected to vary between the hydraulic models and are

supposed fixed. Consequently, only the scale parameters for water pressure head, $|h_g|$, is expected to compensate for the very low values of $c_p$ when the vGM and KG models are used. We conclude that the value of $|h_g|$ must tends towards infinite when the shape index tends towards zero for these two models. For KG models, such relation $|h_{KG}|\, x_{KG}$ is related to the relation $|h_{KG}|\, \sigma_{KG}$ since $x_{KG} = (1 + \sigma_{KG})^{-1}$. Our statements imply that very larges values of $\sigma_{KG}(x_{KG} \to 0)$ should be associated to very larges values of $|h_{KG}|$, i.e., to very small pore radius. In other words, fine (coarse) soils with small (large) pores

should have broad (narrow) pore size distributions. These considerations are in line the previous studies on KG by Pollacco et al. (2013) (see Fig. 1 of their paper), and Fernández-Gálvez et al.. These authors even related the pore radius of soil to the standard deviation of pore size distribution with a strongly decreasing function. Similar trends relating to some extent a relation between pore mean and pore standard deviation should also apply to vGM model given our findings (and to avoid null sorptivity for soils with broad pore size distributions). More investigations are needed to verify for real soils such link between

average pore size and standard deviations. More investigations are also needed to guide on the proper choice of the hydraulic models as a function of the type of soil, as already suggested by Fuentes et al. (1992). These aspects will be the subject of a specific study.

Regarding the numerical accuracy of computed values of $c_p$, we used analytical formulations for the BC, vGB and vGM models as detailed in Eqs. (53). These values are expected to be perfect without any error since their correspond to the ap-

plication of exact analytical formulations. Instead, we used the mixed numerical formulation defined by Eqs. (53) for the KG model that relies on the numerical integration of the hydraulic conductivity or diffusivity functions. In that case, the numerical integration may bring some numerical errors. The mixed form Eq. (47) was designed to minimize numerical indetermination and uncertainty. Such formulation was applied to the other models (BC, vGB and vGM) and the resulting values were compared against the analytical formulations (considered as the benchmark). A perfect agreement was obtained (errors $< 1\%$), thus

validating the numerical mixed formulation and making the authors very confident on the values tabulated in Table 1. Note that the promotion of the numerical mixed formulation and the study of its uncertainty will be the subject of another study.

## 3.3 Upscaling sorptivity $S_K^2\,(h_0, 0)$ from $c_p$

In this section, we elaborate on the use of Eq. (22) for the easy and straightforward computation of $S_K^2\,(h_0, 0)$ from the tabulated values of $c_p$ (Table 1). The proposed scaling procedure Eq. (22) allows the computation of $S_K^2\,(h_0, 0)$ given initial

hydric conditions (water contents or the water pressure heads), hydraulic shape and scale parameters, and specific hydraulic models selected among the studied hydraulic models Eqs. (7)-(10):

1. Use the shape parameter ($\lambda_{BC}$, $m_{vGB}$, $m_{vGM}$ or $\sigma_{KG}$ to compute the related shape index, $x$, considering the following definitions: $x_{vGB} = m_{vGB}$, $x_{vGM} = m_{vGM}$, $x_{KG} = 1/(1 + \sigma_{KG})$, and $x_{BC} = \frac{\lambda_{BC}}{2 + \lambda_{BC}}$.

2. Choose in Table 1 the value of $c_p$ corresponding to the shape index, $x$, and the chosen WRHC functions.

3. Consider or compute the initial water content $\theta_0$ or $\theta\,(h_0)$ depending on the observation for the description of the initial condition (either $\theta_0$ or $h_0$).





4. Compute the related hydraulic conductivity, $K_0 = K(\theta_0)$, using the HC function.

5. Compute the correcting factors $R_\theta = \frac{\theta_s - \theta_0}{\theta_s - \theta_r}$ and $R_K = \frac{K_s - K_0}{K_s}$.

6. Compute the scaled air-entry water pressure head: $|h_a^*| = |\frac{h_a}{h_g}|$

7. Compute the square scaled sorptivity $S_K^{2*}(h_0^*, 0) = R_K R_\theta (c_p - 2|h_a^*|) + 2R_\theta |h_a^*|$

8. Upscale to derive the square sorptivity: $S_K^2(h_0, 0) = S_K^{2*}(h_0^*, 0)(\theta_s - \theta_r)K_s|h_g|$

As an illustrative example, let consider the case of a loamy soil submitted to water saturation with a slightly positive water pressure head at the surface ($h_1 = 0$) and an initial water pressure head of $h_0 = -10$ m (dry conditions). The loamy soil has the features of "loam" as defined in the database of Carsel and Parrish (1988). Its WRHC functions are described by the vGM model, with the following shape and scale parameters: $\theta_r = 0.078$, $\theta_s = 0.43$, $h_g = -277$ mm, and $K_s = 2.88 \ 10^{-3}$ mm s$^{-1}$, $n_{vGM} = 1.56$, and $l_{vGM} = 0.5$. The application of the step-by-step procedure gives the following results:

1. Shape index: $x = m_{vGM} = 1 - 1/n_{vGM}$ leading to $x = 0.359$

2. Corresponding value of $c_p$: $c_p = 0.480$, given Table 1 ("vGM model" column, $x = 0.36$)

3. Initial water content: computed from the initial water pressure head of -10 m using vGM-WR function, i.e., Eqs. (9):
   $\theta_0 = 0.125$.

4. Initial hydraulic conductivity: computed from the initial water content using vGM-HC function, i.e., Eqs. (9):
   $K_0 = 1.87 \ 10^{-9}$ mm s$^{-1}$.

5. Corresponding correction factors: $R_\theta = \frac{\theta_s - \theta_0}{\theta_s - \theta_r}$ and $R_K = \frac{K_s - K_0}{K_0}$, leading to:
   $R_\theta = 0.865$ and $R_K = 1.000$

6. Air-entry water pressure head: no air-entry water pressure head, consequently, $|h_a^*| = 0$

7. Square scaled sorptivity: $S_K^{2*}(h_0^*, 0) = R_K R_\theta c_p$ leading to:
   $S_K^{2*}(h_0^*, 0) = 0.416$

8. Sorptivity: $S_K^2(h_0, 0) = S_K^{2*}(h_0^*, 0)(\theta_s - \theta_r)K_s|h_g|$, leading to:
   $S_K^2(h_0, 0) = 0.117$ mm$^2$ s$^{-1}$, and $S_K(h_0, 0) = 0.342$ mm s$^{-\frac{1}{2}}$

To check the accuracy of the proposed approximation, we computed the nominal value of the sorptivity, using the regular Eq. (2). We found a very close value, with less than $0.5\%$ relative error, demonstrating the accuracy of the proposed scaling procedure Eq. (22).

As a second illustrative example, we consider the computation of sorptivity for the case of BC model, for the same conditions. The difference with the previous case is that the BC model has a non-null air-entry water pressure head, inducing a non-null





saturated sorptivity. We consider the same loamy soil with the following parameters for BC model: $_r = 0.078$, $\theta_s = 0.43$, $h_g = -277$ mm, and $K_s = 2.88 \, 10^{-3}$ mm s$^{-1}$, with a value of $\lambda_{BC} = 0.56$. $\lambda_{BC}$ was deduced from the previous value of $n = 1.56$ considering the usual relation $\lambda = mn$, as suggested by Haverkamp et al. (2005). The application of the proposed procedure leads to the following computations:

1. Shape index: $x_{BC} = \lambda_{BC}/(2 + \lambda_{BC})$ leading to a value of $x_{BC} = 0.219$

2. Corresponding value of $c_p$: $c_p = 2.678$ (see Table 1, "BC model" column for $x_{BC} = 0.22$)

3. Initial water content: computed from the initial water pressure head of -10 m using BC-WR function, i.e., Eqs. (7): $\theta_0 = 0.125$.

4. Initial hydraulic conductivity: computed from the initial water content using BC-HC function, i.e., Eqs. (7): $K_0 = 3.342 \, 10^{-9}$ mm s$^{-1}$.

5. Corresponding correction factors: $R_\theta = \frac{\theta_s - \theta_0}{\theta_s - \theta_r}$ and $R_K = \frac{K_s - K_0}{K_0}$, leading to: $R_\theta = 0.866$ and $R_K = 1.000$

6. Air-entry water pressure head: significant air-entry water pressure head, with, $|h_{BC} = h_a|$ and $|h_a^*| = 1$

7. Square scaled sorptivity: $S_K^{2*}(h_0^*, 0) = R_K R_\theta (c_p - 2|h_a^*|) + -2R_\theta |h_a^*|$ leading to: $S_K^{2*}(h_0^*, 0) = 2.318$

8. Sorptivity: $S_K^2(h_0, 0) = S_K^{2*}(h_0^*, 0)(\theta_s - \theta_r)K_s|h_g|$, leading to: $S_K^2(h_0, 0) = 0.651 \, \text{mm}^2 \, \text{s}^{-1}$, and $S_K(h_0, 0) = 0.806$ mm s$^{-\frac{1}{2}}$

Again, the exact value of sorptivity was estimated using the accurate Eq. (3) and lead to a similar value with a relative error of 1‰. Note that, in this case, due to the non-null air-entry water pressure head, Eq. (3) must be employed instead of Eq. (2) for the determination of the targeted value of sorptivity. The two preceding applications illustrated the accuracy of the proposed scaling procedure Eq. (20) for the two cases of hydraulic functions with and without air-entry water pressure heads. Equation (20) proved appropriate and very accurate for the determination of the sorptivity, $S_K^2(h_0, 0)$.

It must be noted that the proposed scaling procedure applies only for dry initial state. Indeed, Haverkamp et al. (2005) stated that their approximation Eq. (15) was ensured only when $\theta_0 \leq \frac{1}{4}\theta_s$. For fine soils, even a small initial water pressure head may cause $\theta_0 > \frac{1}{4}\theta_s$, which may spoil the proposed scaling procedure. To illustrate this point, we investigated the case of the silty clay soil, as defined by Carsel and Parrish (1988). This soil is defined for the following parameters: $\theta_r = 0.07$, $\theta_s = 0.36$, $h_g = -2000$ mm, and $K_s = 5.555 \, 10^{-5}$ mm s$^{-1}$, $n = 1.09$, and $l = 0.5$. Considering the same value for the initial water pressure head, i.e., $h_0 = -10$ m, the initial water content is $\theta_0 = 0.318$; which exceeds $\frac{1}{4}\theta_s$. The application of the scaling procedure lead to an estimated sorptivity of $0.0475$ mm s$^{-\frac{1}{2}}$, whereas the targeted sorptivity computed with Eq. (2) was $0.0127$ mm s$^{-\frac{1}{2}}$. Such error corresponds to an overestimation by a factor of 2.73. Thus, we advise that the user verify that $\theta_0 \leq \frac{1}{4}\theta_s$ before using the proposed scaling procedure.





## 4  Conclusions

The proper estimation of sorptivity is crucial to understand and model water infiltration into soils. However, its estimation
may be complicated, requiring complicated algebraic derivations and exhibiting potential numerical shortcomings when using
Eq. (2) or Eq. (3). In this study, we present a new scaling procedure for simplifying the computation of sorptivity for the case

of zero water pressure head imposed at surface and dry initial state ($\theta_0 \leq \frac{1}{4}\theta_s$). We based our approach on the combination and
adaptation of the scaling procedure proposed by Ross et al. (1996) and the approximation proposed by Haverkamp et al. (2005).
We then obtain a simple relation that relates the square sorptivity to the product of the square scaled sorptivity, referred to as
$c_p$, the product of scale parameters and two correction factors that account for the initial conditions, (i.e., initial water content
and hydraulic conductivity). The value of the square scaled sorptivity $c_p$ was computed either analytically, when feasible, or

numerically, for four famous sets of hydraulic models: Brooks and Corey, van Genuchten – Mualem, van-Genuchten – Burdine
and Kosugi models. The values of $c_p$ were tabulated as function of specific shape indexes representing similar states of WR
functions (well-graded versus stepwise shapes) between hydraulic models. The proposed scaling procedure is very easy of use.
Once a given hydraulic model is selected with related shape and scale parameters, the procedure steps are easy to perform:
computation of the shape index from the shape parameters, reading of the corresponding value of $c_p$ in Table 1, computation

of the correction factors (ratios in hydraulic conductivity and water contents, $R_K$ and $R_\theta$), computation of the square scaled
sorptivity from $c_p$ and these correction factors, and, lastly, upscaling by multiplying with the scale parameters. All these steps
are easy to conduct and straightforward. Illustrative examples are proposed at the end of this study and the accuracy of the
proposed scaling procedure is clearly demonstrated (with errors less than $1\%$), provided that the initial water content fulfills
the conditions: ($\theta_0 \leq \frac{1}{4}\theta_s$).

535       In addition to providing a straightforward method for the determination of sorptivity, this study brings very interesting
findings on the square scaled sorptivity $c_p$ and its dependency upon the shape index, $x$ and the chosen hydraulic models.
The results show that the function $c_p(x)$ strongly depends on the hydraulic model selected for the WRHC curves. If all the
functions $c_p(x)$ converge for the same value, i.e., 2, close to $x = 1$ (stepwise WR functions – narrow pore size distribution),
they strongly divert close to $x = 0$ (graded WR functions – broad pore size distribution), with values of  0 for vGM and KG

models versus 3 - 4 for the vGB and BC models. However, the sorptivity should remain the same regardless of the selected
hydraulic model: one soil submitted to peculiar initial conditions, one single sorptivity. Consequently, the contrast of scaled
sorptivity must be compensated by a contrast in scale parameters. However, among scale parameters, the residual and saturated
water contents and the saturated hydraulic conductivity cannot be changed between models, since they characterize the dry and
saturated states of the same soil. Consequently, the value of the scale parameter $h_g$ must be the one to compensate. Previous

studies on the Kosugi model have already hypothesized a strong relation between the scale parameter $h_{KG}$ and the standard
deviation $\sigma_{KG}$ (Pollacco et al., 2013). In other words, the scale parameter $h_{KG}$ should be parametrized as a function of the
shape parameter $\sigma_{KG}$, to get plausible WRHC functions and estimates of sorptivity. We may also expect the same link between
the scale parameter $h_{g,vGM}$ and the shape parameter $m_{vGM}$ to avoid unphysical scenarios and null sorptivity. However, such
hypothesis has never been suggested and requires further investigations. These results show the need to better understand the





mathematical properties of the hydraulic models, including the links between shape and scale parameters, and to better relate these properties to the physical processes of water infiltration into soils (Fuentes et al., 1992).

In addition to the proposed scaling procedure, this study gave the opportunity to derive analytically the scaled sorptivity for the three models, BC, vGB and vGM, thus confirming the expressions provided by previous studies. For the vGM model, the analytical derivation is brand new and had never been proposed before. Its use if of great interest and could be implemented

into soil hydraulic characterization methods. For instance, additional BEST methods could be developed ob the basis of the use of the proposed formulations for square scaled sorptivity to relate sorptivity to shape and scale parameters. In more details, the prior estimation of shape parameters allows the determination of the parameter $c_p$ using Eq. (45). Then, the estimation of saturated hydraulic conductivity, and sorptivity allows the determination of scale parameter $h_g$ once $c_p$ is determined. A similar procedure may be proposed for the vGM model, using the Eq. (46) that defines the parameter $c_p$ to the shape parameter $m_{vGM}$.

The development of BEST method for the specific vGM hydraulic model, that is much more used than the vGB model, will be the subject of further investigations. It would also be interesting to derive somehow the residual water content, and not to assume it to be equal to zero as it might alter the shape of the soil water retention function. The use of the scaled sorptivity for these purposes are the subject of ongoing studies.

*Code availability.* Note all computations were done using Scilab free software. The scripts for the computation of Eqs. (25)-(28) for the

computation of WRHC functions, Eqs. (35)-(38) for the computation of the dimensionless diffusivity, and Eqs. (35)-(38) for the computation of the $c_p$ parameter can be downloaded online: https://zenodo.org/record/4587160 (Lassabatere, 2021).

**Appendix A:  Dimensionless hydraulic diffusivity functions, $D^*\left(S_e\right)$**

In the appendices, for the sake of clarity the notations of the shape parameters were simplified to $\lambda$, $m$, $n$, $\sigma$, in order to avoid heavy equations. The dimensionless diffusivity functions were derived from their definition $D^*\left(S_e\right)$, applying $D^*\left(S_e\right) =$

$K_r\left(S_e\right)\frac{dh^*}{dS_e}$. This task requires first to derive the inverse functions for the dimensionless water retention curves. The following equations can be easily found through usual algebraic developments:

$$h_{BC}^*\left(S_e\right) = -S_e^{-\frac{1}{\lambda}} \tag{A1}$$

$$h_{vGB}^*\left(S_e\right) = -\left(S_e^{-\frac{1}{m}}-1\right)^{\frac{1}{n}} \quad \text{with} \quad m = 1-\frac{2}{n} \tag{A2}$$

$$h_{vGM}^*\left(S_e\right) = -\left(S_e^{-\frac{1}{m}}-1\right)^{\frac{1}{n}} \quad \text{with} \quad m = 1-\frac{1}{n} \tag{A3}$$

$$h_{KG}^*\left(S_e\right) = -e^{\sqrt{2}\sigma\,erfc^{-1}\left(2\,S_e\right)} \tag{A4}$$





where $erfc^{-1}$ is the inverse function of the complementary error function. These functions can be differentiated to define their relative derivatives, $\frac{dh^*}{dS_e}$:

$$\frac{dh^*_{BC}}{dS_e}(S_e) = \frac{1}{\lambda} S_e^{-\frac{1}{\lambda}-1} \tag{A5}$$

$$\frac{dh^*_{vGB}}{dS_e}(S_e) = \frac{1-m}{2m} S_e^{-\frac{1+m}{2m}} \left(1 - S_e^{\frac{1}{m}}\right)^{-\frac{m+1}{2}} \tag{A6}$$

$$\frac{dh^*_{vGM}}{dS_e}(S_e) = \frac{1-m}{m} S_e^{-\frac{1}{m}} \left(1 - S_e^{\frac{1}{m}}\right)^{-m} \tag{A7}$$

$$\frac{dh^*_{KG}}{dS_e}(S_e) = \sqrt{2\pi}\sigma e^{\left(erfc^{-1}(2S_e)\right)^2 + \sqrt{2}\sigma erfc^{-1}(2S_e)} \tag{A8}$$

The differentiation of the function $h^*_{KG}$ involves the following usual rules of differentiation $(f \circ g)' = f' \circ g \cdot g'$ and $(f^{-1})' = \frac{1}{f' \circ f^{-1}}$, considering bijective functions. We also use the usual derivative of the function $erf$ function, $erf'(x) = \frac{2}{\sqrt{\pi}}e^{-x^2}$, and the relation between $erfc$ and $erf$ functions, $erfc(x) = 1 - erf(x)$.

The derivatives $\frac{dh^*}{dS_e}$ can now be multiplied with the hydraulic conductivity:

$$K_{r,BC}(S_e) = S_e^\eta \tag{A9}$$

$$K_{r,vGB}(S_e) = S_e^\eta \tag{A10}$$

$$K_{r,vGM}(S_e) = S_e^l \left(1 - 2\left(1 - S_e^{\frac{1}{m}}\right)^m + \left(1 - S_e^{\frac{1}{m}}\right)^{2m}\right) \tag{A11}$$

$$K_{r,KG}(S_e) = S_e^l \left(\frac{1}{2}erfc\left(erfc^{-1}(2S_e) + \frac{\sigma}{2}\right)\right)^2 \tag{A12}$$

Note that for the hydraulic conductivity function of vGM model, $K_{r,vGM}$, we distributed the terms according to $(a+b)^2 = a^2 + 2ab + b^2$. The multiplication of Eqs. (A5)-(A8) with the expressions of relative conductivity Eq. (A9)-(A12) lead to the following expressions of dimensionless diffusivity:

$$\begin{cases} D^*_{BC}(S_e) = \frac{1}{\lambda} S_e^{\eta - \left(\frac{1}{\lambda}+1\right)} \\ D^*_{vGB}(S_e) = \frac{1-m}{2m} S_e^{\eta - \frac{1+m}{2m}} \left(1 - S_e^{\frac{1}{m}}\right)^{-\frac{1+m}{2}} \\ D^*_{vGM}(S_e) = \frac{1-m}{m} S_e^{l-\frac{1}{m}} \left(\left(1 - S_e^{\frac{1}{m}}\right)^{-m} + \left(1 - S_e^{\frac{1}{m}}\right)^m - 2\right) \\ D^*_{KG}(S_e) = \frac{1}{2}\sqrt{\frac{\pi}{2}}\sigma S_e^l \left(erfc\left(erfc^{-1}(2S_e) + \frac{\sigma}{\sqrt{2}}\right)\right)^2 e^{\left(erfc^{-1}(2S_e)\right)^2 + \sqrt{2}\sigma erfc^{-1}(2S_e)} \end{cases} \tag{A13}$$

These equations, Eqs. (A13), correspond to the expressions of Eqs. (30)-(33). Afterwards, the combination with the capillarity model Eq. (34) leads to Eqs. (35)-(38).





### Appendix B: Analytical developments for $c_p$ parameter

#### B1    Parameter $c_p$ for BC model

For the BC model, we need to account for the air entry pressure, $h_a^* = -1$. We remind that by convention, the scale parameters for water pressure head, $h_{BC}$ is equalled to the air-entry water pressure head, $h_a$. We then use the equation Eq. (23) with $|h_a^*| =$

1. Then, the first part $\int_0^1 (1 + S_e) D_{BC}^* (S_e) dS_e$ can be integrated analytically given that the hydraulic diffusivity $D_{BC}^* (S_e)$ obeys a power law. The following developments can be done:

$$
\begin{aligned}
c_{p,BC} &= \int_0^1 (1 + S_e) D_{BC}^* (S_e) dS_e + 2 |h_a^*| \\
&= \int_0^1 (1 + S_e) D_{BC}^* (S_e) dS_e + 2
\end{aligned}
\tag{B1}
$$

$$
\begin{aligned}
\int_0^1 (1 + S_e) D_{BC}^* (S_e) dS_e &= \int_0^1 D_{BC}^* (S_e) dS_e + \int_0^1 S_e D_{BC}^* (S_e) dS_e \\
&= \int_0^1 \frac{1}{\lambda} S_e^{\eta - \left(\frac{1}{\lambda} + 1\right)} dS_e + \int_0^1 \frac{1}{\lambda} S_e S_e^{\eta - \left(\frac{1}{\lambda} + 1\right)} dS_e \\
&= \frac{1}{\lambda} \left( \int_0^1 S_e^{\eta - \left(\frac{1}{\lambda} + 1\right)} dS_e + \int_0^1 S_e^{\eta - \frac{1}{\lambda}} dS_e \right) \\
&= \frac{1}{\lambda} \left( \frac{1}{\eta - \frac{1}{\lambda}} + \frac{1}{\eta - \frac{1}{\lambda} + 1} \right) \\
&= \frac{1}{\eta \lambda - 1} + \frac{1}{\eta \lambda + \lambda - 1}
\end{aligned}
\tag{B2}
$$

The final expression can be easily computed by combining Eqs. (B1) and Eqs. (B2):

$$
c_{p,BC} (\lambda, \eta) = \frac{1}{\eta \lambda - 1} + \frac{1}{\eta \lambda + \lambda - 1} + 2
\tag{B3}
$$

The concatenation of the Eq. (B3) with the capillary model, Eq.( 34), leads to the following final expression:

$$
c_{p,BC} (\lambda) = 2 + \frac{1}{3\lambda + 1} + \frac{1}{4\lambda + 1}
\tag{B4}
$$

These development demonstrate the equations proposed for $c_p$ for the BC model, i.e., Eqs. (40)-(41). This demonstration is in
line with Varado et al. (2006).


## B2 Parameter $c_p$ for vGB model

For vGB model, and the remaining models, there is no air-entry water pressure head, $h_a^* = 0$, leading to:

$$c_{p,vGB} = \int_0^1 (1 + S_e)\, D_{vGB}^*(S_e)\, dS_e + 2\,|h_a^*|$$

$$= \int_0^1 (1 + S_e)\, D_{vGB}^*(S_e)\, dS_e \tag{B5}$$

Then, the integral $\int_0^1 (1 + S_e)\, D_{vGB}^*(S_e)\, dS_e$ can be decomposed into well-known integrals:

$$c_{p,vGB}(m,\eta) = \int_0^1 (1 + S_e)\, D_{vGB}^*(S_e)\, dS_e$$

$$= \int_0^1 D_{vGB}^*(S_e)\, dS_e + \int_0^1 S_e\, D_{vGB}^*(S_e)\, dS_e$$

$$= \int_0^1 \frac{1-m}{2m} S_e^{\eta - \frac{1+m}{2m}} \left(1 - S_e^{\frac{1}{m}}\right)^{-\frac{1+m}{2}} dS_e + \int_0^1 S_e \frac{1-m}{2m} S_e^{\eta - \frac{1+m}{2m}} \left(1 - S_e^{\frac{1}{m}}\right)^{-\frac{1+m}{2}} dS_e$$

$$= \frac{1-m}{2m} \left( \int_0^1 S_e^{\eta - \frac{1+m}{2m}} \left(1 - S_e^{\frac{1}{m}}\right)^{-\frac{1+m}{2}} dS_e + \int_0^1 S_e\, S_e^{\eta - \frac{1+m}{2m}} \left(1 - S_e^{\frac{1}{m}}\right)^{-\frac{1+m}{2}} dS_e \right)$$

$$\frac{1-m}{2m} \left( \int_0^1 S_e^{\eta - \frac{1+m}{2m}} \left(1 - S_e^{\frac{1}{m}}\right)^{-\frac{1+m}{2}} dS_e + \int_0^1 S_e^{\eta + 1 - \frac{1+m}{2m}} \left(1 - S_e^{\frac{1}{m}}\right)^{-\frac{1+m}{2}} dS_e \right)$$

$$= \frac{1-m}{2m} \left( \int_0^1 \left(S_e^{\frac{1}{m}}\right)^{m\eta - \frac{1+m}{2}} \left(1 - S_e^{\frac{1}{m}}\right)^{-\frac{1+m}{2}} dS_e + \int_0^1 \left(S_e^{\frac{1}{m}}\right)^{m\eta + m - \frac{1+m}{2}} \left(1 - S_e^{\frac{1}{m}}\right)^{-\frac{1+m}{2}} dS_e \right) \tag{B6}$$

The change of variable $y = S_e^{\frac{1}{m}}$ provides the following expressions:

$$c_{p,vGB}(m,\eta) = \frac{1-m}{2} \int_0^1 y^{m\eta + \frac{m}{2} - \frac{3}{2}} (1-y)^{-\frac{m+1}{2}} dy + \frac{1-m}{2} \int_0^1 y^{m\eta + \frac{3m}{2} - \frac{3}{2}} (1-y)^{-\frac{m+1}{2}} dy \tag{B7}$$

We can recognize in equation the beta function B and use its following properties (assuming $x > 0$ and $y > 0$):

$$B(x,y) = \int_0^1 t^{x-1} t^{y-1} dt$$

$$= \frac{\Gamma(x)\,\Gamma(y)}{\Gamma(x+y)} \tag{B8}$$





Where the $\Gamma$ function is already defined by Eq. (43): $\Gamma(z) = \int_0^{+\infty} t^{z-1} e^{-t} dt$ $(z > 0)$. Then, the parameter $c_{p,vGB}$ can be expressed as follows:

$$
\begin{aligned}
c_{p,vGB}(m,\eta) &= \frac{1-m}{2} B\left(m\eta + \frac{m}{2} - \frac{1}{2}, \frac{1-m}{2}\right) + \frac{1}{n} B\left(m\eta + \frac{3m}{2} - \frac{1}{2}, \frac{1-m}{2}\right) \\
&= \frac{1-m}{2} \Gamma\left(\frac{1-m}{2}\right) \left(\frac{\Gamma\left(m\eta + \frac{m}{2} - \frac{1}{2}\right)}{\Gamma(m\eta)} + \frac{\Gamma\left(m\eta + \frac{3m}{2} - \frac{1}{2}\right)}{\Gamma(m\eta + m)}\right) \\
&= \Gamma\left(\frac{3-m}{2}\right) \left(\frac{\Gamma\left(m\eta + \frac{m}{2} - \frac{1}{2}\right)}{\Gamma(m\eta)} + \frac{\Gamma\left(m\eta + \frac{3m}{2} - \frac{1}{2}\right)}{\Gamma(m\eta + m)}\right)
\end{aligned}
\tag{B9}
$$

The last equation uses the fact that $\Gamma(z+1) = z\Gamma(z)$. The Eq. (B9) corresponds to the equation suggested by Haverkamp et al. (2005), considering $m = 1 - \frac{2}{n}$:

$$
c_{p,vGB}(n,m,\eta) = \Gamma\left(1 + \frac{1}{n}\right) \left(\frac{\Gamma\left(m\eta - \frac{1}{n}\right)}{\Gamma(m\eta)} + \frac{\Gamma\left(m\eta + m - \frac{1}{n}\right)}{\Gamma(m\eta + m)}\right)
\tag{B10}
$$

Note that the proposed simplification using the beta function, Eq. (B9) requires that $m\eta + \frac{m}{2} - \frac{1}{2} > 0$, which is quite evident since $m < 1$. When Eq. (B9) is combined with capillary model, i.e., Eq. (34), the expression becomes:

$$
c_{p,vGB}(m) = \Gamma\left(\frac{3-m}{2}\right) \left(\frac{\Gamma\left(\frac{1+5m}{2}\right)}{\Gamma(1+2m)} + \frac{\Gamma\left(\frac{1+7m}{2}\right)}{\Gamma(1+3m)}\right)
\tag{B11}
$$

The expressions of the parameter $c_p$ for the vGB model are accurately demonstrated, leading to Eqs. (42) and (44).

## B3  Parameter $c_p$ for vGM model

For the vGM model, the same equation Eq. (B5) applies and can be cut into two parts:

$$
\begin{aligned}
c_{p,vGM}(m,l) &= \int_0^1 (1 + S_e) D_{vGM}^*(S_e) dS_e \\
&= \underbrace{\int_0^1 D_{vGM}^*(S_e) dS_e}_{A} + \underbrace{\int_0^1 S_e D_{vGM}^*(S_e) dS_e}_{B}
\end{aligned}
\tag{B12}
$$

For the sake of clarity, we demonstrate separately the simplifications of the two terms A and B:

$$
\begin{aligned}
A &= \frac{1-m}{m} \int_0^1 S_e^{l - \frac{1}{m}} \left(\left(1 - S_e^{\frac{1}{m}}\right)^{-m} + \left(1 - S_e^{\frac{1}{m}}\right)^m - 2\right) dS_e \\
&= \frac{1-m}{m} \left(\int_0^1 S_e^{l - \frac{1}{m}} \left(1 - S_e^{\frac{1}{m}}\right)^{-m} dS_e + \int_0^1 S_e^{l - \frac{1}{m}} \left(1 - S_e^{\frac{1}{m}}\right)^m dS_e - 2\int_0^1 S_e^{l - \frac{1}{m}} dS_e\right) \\
&= \frac{1-m}{m} \left(\int_0^1 \left(S_e^{\frac{1}{m}}\right)^{ml-1} \left(1 - S_e^{\frac{1}{m}}\right)^{-m} dS_e + \int_0^1 \left(S_e^{\frac{1}{m}}\right)^{ml-1} \left(1 - S_e^{\frac{1}{m}}\right)^m dS_e - 2\int_0^1 S_e^{l - \frac{1}{m}} dS_e\right)
\end{aligned}
\tag{B13}
$$





The last term of A can be simplified easily:

$$
\int_0^1 S_e^{l-\frac{1}{m}} dS_e = \int_0^1 S_e^{l-\frac{1}{m}} dS_e = \left[ \frac{S_e^{l-\frac{1}{m}+1}}{l-\frac{1}{m}+1} \right]_0^1
$$

$$
= \frac{m}{(l+1)m-1} \tag{B14}
$$

For the two first, terms, we use the same change of variable $y = S_e^{\frac{1}{m}}$ as above to transform the integrals, leading to:

$$
A = (1-m) \left( \int_0^1 y^{ml+m-2} (1-y)^{-m} dy + \int_0^1 y^{ml+m-2} (1-y)^m dy - \frac{2}{(l+1)m-1} \right) \tag{B15}
$$

In this case, we need to assume that $ml + m - 2 > -1$, i.e. $m > \frac{1}{l+1}$ to use the beta function. Such a condition corresponds to

$m > \frac{2}{3}$, for the by-default value of $l = \frac{1}{2}$. The two first integrals can then be replaced using the beta and the gamma functions, leading to the final expression for part A of $c_p$:

$$
A = (1-m) \left( B(m(l+1)-1, 1-m) + B(m(l+1)-1, 1+m) - \frac{2}{(l+1)m-1} \right)
$$

$$
= (1-m) \left( \frac{\Gamma(m(l+1)-1)\Gamma(1-m)}{\Gamma(ml)} + \frac{\Gamma(m(l+1)-1)\Gamma(1+m)}{\Gamma(m(l+2))} - \frac{2}{(l+1)m-1} \right) \tag{B16}
$$

By analogy, the following developments come out for parameter B:

$$
B = (1-m) \left( \frac{\Gamma(m(l+2)-1)\Gamma(1-m)}{\Gamma(ml+m)} + \frac{\Gamma(m(l+2)-1)\Gamma(1+m)}{\Gamma(m(l+3))} - \frac{2}{(l+2)m-1} \right) \tag{B17}
$$

The simplification for B is valid as soon as $m > \frac{1}{2+l}$, which is the case since we suppose that $m > \frac{1}{1+l}$. After rearranging terms, the following expressions comes out for the dimensionless sorptivity:

$$
c_{p,vGM}(m,l) = \Gamma(2-m) \left( \frac{\Gamma(m(l+1)-1)}{\Gamma(ml)} + \frac{\Gamma(m(l+2)-1)\Gamma(1-m)}{\Gamma(ml+m)} \right)
$$
$$
+ (1-m) \left[ \left( \frac{\Gamma(m(l+1)-1)\Gamma(1+m)}{\Gamma(m(l+2))} + \frac{\Gamma(m(l+2)-1)\Gamma(1+m)}{\Gamma(m(l+3))} \right) - 2 \left( \frac{1}{(l+1)m-1} + \frac{1}{(l+2)m-1} \right) \right] \tag{B18}
$$

$$
c_{p,vGM}(m,l) = \Gamma(2-m) \left( \frac{\Gamma(m(l+1))}{(m(l+1)-1)\Gamma(ml)} + \frac{\Gamma(m(l+2))}{(m(l+2)-1)\Gamma(ml+m)} \right)
$$
$$
+ (1-m) \left[ \left( \frac{\Gamma(m(l+1)-1)\Gamma(1+m)}{\Gamma(m(l+2))} + \frac{\Gamma(m(l+2)-1)\Gamma(1+m)}{\Gamma(m(l+3))} \right) - 2 \left( \frac{1}{(l+1)m-1} + \frac{1}{(l+2)m-1} \right) \right] \tag{B19}
$$

Note that, as stated above, that equation theoretically should apply only for the case of $m > \frac{1}{1+l}$, which is quite restrictive. However, thanks to the analyticity of the functions involved in the expression, this equality remains valid for $m > \frac{1}{1+l}$ and can be considered for any value of $m \in [0,1]$ provided that $m \neq \frac{1}{1+l}$ and $m \neq \frac{1}{2+l}$. The Eq. (B19) demonstrates the Eqs. (42). The combination of these equations with the capillary model Eq. (34) leads to Eqs. (46).





*Author contributions.* L.L.: established the question, performed the analytical developments, computed the numerical results, provided the

first draft of the manuscript. P.-E.P. performed the analytical developments and verified the numerical results. D.Y., B.L., D.M.-F., S.d.P. and

M.R. verified parts of the developments and numerical computations. J. P. and J. F.-G. helped for the use of the Kosugi model. R.D.S. and

M.A.N. helped for the editing and the presentation of the results, in particular the writing of the application of the proposed approach for

end-users. All the authors contributed to the editing of the manuscript.

*Competing interests.* No competing interest to declare.

*Acknowledgements.* This work was performed within the INFILTRON project supported by the French National Research Agency (ANR-

17-CE04-010).





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
