# Peer review of "Scaling procedure for straightforward computation of sorptivity"

_Hydrology and Earth System Sciences, 2021_

## Author Response (AR1)

**Laurent Lassabatere**
ENTPE, LEHNA IAPHY
3 Rue Maurice Audin
Vaulx-en-Velin, 69120
**Phone**: +33 (0)4-72-04-7057
**E-mail**: laurent.lassabatere@entpe.fr
**URL**: http://www.entpe.fr

17 July 2021

To HESS editor
Nunzio Romano

**Authors' answers to the Reviewers and the Editor.**

Dear Editor,

Many thanks for your positive comments. The authors have revised their manuscript and addressed all the points raised by the reviewers. We hope that the revised manuscript will be considered for publication.

You'll find below the answer to the reviewers' comments. The reviewers' suggestions are in black, and the authors' responses in blue.

**Reviewer 1**

It is a great work where a rescaling procedure is developed to calculate the sorptivity in which, through the calculation of the parameter $c_p$, different models are compared which start from different hypotheses, making these results remarkable. Authors are recommended to add and / or replace the work of Brutsaert (1976) with the work of Parlange (1975). Finally, a general revision is recommended to correct typos.

Dear Reviewer, the authors warmly thank you for the time dedicated to reviewing our paper and his positive feedback. We have checked for typos carefully. In this paper, the authors considered Parlange's equation for the flux concentration function. However, other flux concentration functions may be used in the integral that defines the sorptivity, like those proposed by Crank (1979), Philip and Knight (1974), Parlange (1975), Brutsaert (1976). However, this topic is beyond the frame of the proposed study. We intend to deal with the sensitivity of the computation of sorptivity with regard to the choice of the flux concentration function in a specific paper. We have added some sentences on that point (lines 48-50 of the revised manuscript, marked file). Additional references: Crank, J.: The mathematics of diffusion, Oxford university press., 1979. Philip, J. R. a and Knight, J. H. b: On solving the unsaturated flow equation: 3. new quasi-analytical technique, Soil Science, 117(1), 1–13, 1974.

**Reviewer 2**

This is an excellent paper that provides scaling equations to estimate sorptivity for a wide range of hydraulic functions as well as initial and final soil moisture status. The mathematical derivation is thorough and accurate to the best of what I was able to follow. I have two main comments and a few minor corrections.

Dear Reviewer, The authors thank you for your very positive review and the time dedicated to reviewing the paper.

Regarding the two main comments, we revised the paper to make it clear that sorptivity has two parts, the unsaturated sorptivity that corresponds to $c_p$' and the saturated part that corresponds to $2|h_a^*|$. The saturated part is related to the air-entry water pressure heads $|h_a^*|$ that equals unity only for the Delta and the BC functions. In that case, the saturated part equals $2 \cdot |h_a^*| = 2$. Note that by convention, for the Delta and BC functions, the scale water pressure head, $h_g$, is taken equal to the air entry pressure head, so that $|h_a^*| = 1$. For the other hydraulic functions, $|h_a^*| = 0$, and the saturated part of sorptivity is null. We implemented the reviewer's suggestions on discussing both parts of the sorptivity as a function of the shape indexes and revised Figure 3.

**Comment 1 by Reviewer RC2.** Eq. (22) gives rise to contrasting values of sorptivity for the different hydraulic conductivity functions. The authors attribute this difference to the dependence of the parameter $c_p$ on the hydraulic functions (see section 4.4). However, sorptivity as defined in Eq 22 also varies with $|h_g|$ and $2|h_a^*|$. Indeed, the authors defined a variable $c_p' = c_p - 2|h_a^*|$. Therefore, consider deriving shape indices for $c_p'$.

In the result section (section 3.2), we plotted the scaled dimensionless sorptivity, $c_p$, as a function of shape indexes (see revised version of the manuscript). For the Delta and BC functions, the scaled sorptivity lumps the saturated parts, equal to $2|h_a^*| = 2$ plus the unsaturated part, $c_p$'. For the other functions, the scaled sorptivity corresponds directly to the unsaturated part, $c_p$', since $|h_a^*|$ and the saturated part of sorptivity are null. We have added some sentences on this point and discussed the evolution of the unsaturated part $c_p$' as a function of the shape index, as suggested by the reviewer (lines 430-435 of the revised manuscript, marked file).

**Comment 2 by Reviewer RC2.** What is the value of $h_a$? I suspect it is equal to $h_g$ for the Delta and BC models and zero for the others. If that is the case, $|h_a^*| = 1$ for the former two and 0 for the others (see the top of Page 5). Thus, $c_p' = c_p - 1$ or $c_p' = c_p$. If you plot $c_p'$, the curves for $c_{p,d}'$ and $c_{p,BC}'$ in Figure 3 would be lowered by 1 and the in (a) and (c). This would reduce the dissimilarity between

the various hydraulic functions a bit.

The reviewer is correct to state that $|h_a^*|$ = 1 by convention for the Delta and BC functions and zero for the other hydraulic functions. Thus, the unsaturated part of the sorptivity equals $c_p' = c_p$–2 for the Delta and BC functions and $c_p' = c_p$ for the others. We have added $c_p'$ in Figure 3 and discussed this point in the revised version of the paper. In addition, we properly defined the two parameters $c_p$ and $c_p$' and clarified their links (lines 188-196 of the revised manuscript, marked file).

**Answers to the reviewer's suggestions.**

We have carefully searched for typos and revised the manuscript. We rewrote the inappropriate sentences. More importantly, the reviewer suggested changing the structure of section 2.1 with the successive presentations of the dimensional equations and the scaling procedure. The authors have revised this section accordingly.

- Reviewer: In the first line of the introduction, verify if sorptivity is actually used for desorption. Authors: Regarding sorptivity and its definition (line 27 of the revised manuscript, marked file), it corresponds to the description of sorptivity proposed by Minasmy and Cook (2011): "Sorptivity is a measure of the capacity of the medium to absorb or desorb liquid by capillarity." The concept of sorptivity can be considered regarding the two sides of the same coin, i.e., water adsorption and desorption, with potential hysteresis effects.

- Reviewer: Eq (4), Eq (5), and elsewhere there is no need to show the detailed step-by-step derivation of straightforward algebraic manipulations. Authors: We simplified the equations when necessary. However, our goal is to help the reader retrieve all equations and derivations and highlight mathematical developments' main outlines. When the derivations were complicated, we kept only the main steps.

- Reviewer: (i) In the last paragraph of Page 3, rewrite the sentence that starts with "Secondly, ..." (ii) In the same paragraph as above, define "BEST." (iii) In the same paragraph as above, introduce hydraulic functions starting with the delta function to be consistent with how the equations are presented. Authors: the paragraph was rewritten and those points addressed (lines 66-82 of the revised manuscript, marked file).

- Reviewer: Rewrite equation (6) using the Heaviside function since $H$ is defined underneath and later references use $H$ as well. Authors: Done (lines 85 of the revised manuscript, marked file).

- Reviewer: (i) Postpone the introduction of the scaling parameters section 2.1, where they are

used, (ii) Consider moving Eq (23) (definition of $c_p$) to just after Eq (15), where $c_p$ is initially introduced. Also, provide more information of what assumptions were used by Haverkamp et al. in deriving $c_p$. Authors: We decided to change section 2.1 to present the developments with dimensional data before elaborating on the scaling procedure, as suggested by the reviewer in his main comments. Section 2.1 was utterly rewritten (lines 128-213 of the revised manuscript, marked file).

- Reviewer: Edit the incomplete first sentence of section 2.2.2. Authors: The sentence was rewritten (lines 245-247 of the revised manuscript, marked file).

Best regards

Laurent Lassabatere,
Principal Research Fellow
ICTPE (HDR)